# Rapid complete reconfiguration induced actual active species for industrial hydrogen evolution reaction

Luqi Wang[1], Yixin Hao[1], Liming Deng[1], Feng Hu[1], Sheng Zhao[1], Linlin Li[1] & Shengjie Peng [1] ✉

Rational regulation of electrochemical reconfiguration and exploration of activity origin are important foundations for realizing the optimization of electrocatalyst activity, but rather challenging. Herein, we potentially develop a rapid complete reconfiguration strategy for the heterostructures of $CoC_2O_4$ coated by MXene nanosheets ($CoC_2O_4$@MXene) during the hydrogen evolution reaction (HER) process. The self-assembled $CoC_2O_4$@MXene nanotubular structure has high electronic accessibility and abundant electrolyte diffusion channels, which favor the rapid complete reconfiguration. Such rapid reconfiguration creates new actual catalytic active species of $Co(OH)_2$ transformed from $CoC_2O_4$, which is coupled with MXene to facilitate charge transfer and decrease the free energy of the Volmer step toward fast HER kinetics. The reconfigured components require low overpotentials of 28 and 216 mV at 10 and 1000 mA cm$^{-2}$ in alkaline conditions and decent activity and stability in natural seawater. This work gives new insights for understanding the actual active species formation during HER and opens up a new way toward high-performance electrocatalysts.

Hydrogen fuel with high mass specific energy density and environmental friendliness is an excellent energy carrier, contributing to the realization of carbon neutrality[1–3]. Electrocatalytic water splitting is an economical and efficient method for hydrogen production[4,5]. The hydrogen evolution reaction (HER) as one of the half-reactions of water splitting is an important factor affecting hydrogen production technology, which requires the development of efficient HER catalysts with fast kinetics[6–9]. In this regard, the benchmark Pt catalyst is still considered to be the most efficient commercial electrocatalyst for HER. However, they usually suffer from scarcity, high cost, and long-term instability[10–12]. Thus far, transition metal-based materials have turned to be the promising alternatives to noble metal catalysts because of their lower cost and appropriate electronic structure[13,14]. It is reported that they are prone to involving surface reconfiguration processes through electrochemical activation under complex conditions of catalytic reactions, ultimately exhibiting excellent catalytic

performance[15–17]. Therefore, it is greatly attractive if this electrochemical reconfiguration can be utilized to develop new electrocatalysts. However, reconfigured catalysts usually present multiple components and valence states, which complicate the dynamic evolution process, resulting in extremely impediment to the identification of actual catalytic active species and the understanding of catalytic origins[18,19]. As a result, exploring the dynamic reconfiguration mechanism of catalysts and revealing the actual catalytic active species are beneficial to the accurate design of high-efficiency nonprecious metal-based electrocatalysts.

Typically, oxygen evolution reaction (OER) catalysts would undergo a surface reconfiguration process to form highly active metal oxides or (oxy)hydroxides species under electro-oxidative conditions, which have been confirmed as the actual active phase[20,21]. However, the catalytic surface reconfiguration process to improve the electrocatalytic activity rarely occurs during HER[22,23]. Normally, the HER

[1]Jiangsu Key Laboratory of Electrochemical Energy Storage Technologies, College of Materials Science and Technology, Nanjing University of Aeronautics and Astronautics, Nanjing 210016, China. ✉e-mail: pengshengjie@nuaa.edu.cn

catalysts generally exhibit a near-surface nanoscale reconfiguration, such as a core-shell structure, containing numerous internal inert components[24]. Due to their restrictive surface-active area, the derived catalysts reveal incompletely exploited catalytic activity, resulting in low component utilization[25,26]. Meanwhile, the complex composition of the surface-reconfigured catalysts has greatly hindered the in-depth exploration of catalytic origins[27,28]. Considering these aspects, the rational design of catalysts with special structural and chemical properties to trigger complete reconfiguration can effectively eliminate the above problems. The advantages of the complete reconfiguration lie in the adequate electrolyte contact and loose reconfiguration layer for favoring the solution penetration and mass transport[29–31]. Generally, the addition of a hydrophilic material to the catalyst to improve the wettability between the electrode and the electrolyte can assist in inducing complete reconfiguration of the catalyst[20]. Current studies found that the introduction of MXene improved the wettability of catalyst surface, which could fully react with the electrolyte and achieve preliminary reconfiguration[32–34]. Hence, the rational design of MXene-based catalysts is expected to achieve deep or even complete reconfiguration. Moreover, the loose reconfigured structure facilitates accelerative electrolyte supply and fast gas diffusion during vigorous electrocatalytic reactions, which is in favor of hydrogen production at high current densities[35–37]. Meanwhile, since entire unstable components could be converted into thermodynamically steady species during the complete reconfiguration process, it can also realize outstanding catalysis durability even under terrible conditions of natural seawater[38,39]. Therefore, engineering on the complete reconfiguration and the comprehensive understandings are believed to direct purposeful design of high-performance catalysts for industrialization promotion.

Herein, we report a significant conceptual advance of reconfiguration during the HER process based on the $CoC_2O_4$@MXene model catalyst. The obtained heterostructured catalyst features a nanotubular structure, composed of $CoC_2O_4$ nanorods self-assembled with MXene due to the interface-induced effect. This favorable hydrophilic nanotubular structure can immensely increase the penetration of the electrolyte and the ultra-thin MXene with high conductivity provides an effective electronic path, which both facilitates the rapid and deep self-reconfiguration process. In addition, the loose structure is the main factor to drives complete reconfiguration. In-situ characterization techniques and X-ray absorption spectroscopy demonstrate the actual active species of $Co(OH)_2$ transformed from $CoC_2O_4$. The reconfigured components can manipulate the electron delocalization and reduce the energy barrier, leading to fast electron transfer. More importantly, such rapid reconfiguration introduces a highly disordered structure of $Co(OH)_2$ with abundant exposed Co active sites, which can facilitate the adsorption of water molecules and optimize the binding energies of H* intermediates, and thus eventually contributes to the HER kinetics. As a result, the synergistic interaction between $Co(OH)_2$ and MXene ensures remarkable HER activity even at a high current density. Specifically, the reconfigured $CoC_2O_4$@MXene delivers small overpotentials of 28 and 216 mV to reach current densities of 10 and 1000 mA $cm^{-2}$ in 1 M KOH, respectively. Even in alkaline and neutral seawater, the reconfigured $CoC_2O_4$@MXene exhibits superior and stable HER performance. This study not only provides idealized electrocatalysts for highly efficient HER under various conditions but also unprecedentedly reveals the rapid complete reconfiguration during the HER, thereby giving a deep understanding for the performance-enhanced origin and insights into future catalyst design.

## Results

### Synthesis and characterization of morphologies and structures

An interface-induced self-assembly strategy is employed to synthesize the hollow nanotube $CoC_2O_4$@MXene pre-catalyst, which is specifically described in Fig. 1a. Briefly, MXene nanosheets were obtained by selective etching of Al element in MAX phase ($Ti_3AlC_2$) by LiF/HCl acid and the following subsequent exfoliation (Supplementary Fig. 1). After etching and ultrasonic exfoliation treatments, the XRD characteristic peak of $Ti_3AlC_2$ disappears and exhibits a typical peak at 6.1° corresponding to the (002) peak (Supplementary Figs. 2 and 3), demonstrating that $Ti_3AlC_2$ has completely converted to $Ti_3C_2T_x$ MXene[40]. Subsequently, the $CoC_2O_4$@MXene hybrid is finally obtained through dropping oxalic acid dihydrate ($H_2C_2O_4 \cdot 2H_2O$) aqueous solution into the mixture of $Co(NO_3)_2 \cdot 2H_2O$ aqueous solution and MXene nanosheets with continuous stirring under room-temperature. Specifically, $CoC_2O_4$ crystal nuclei are formed on the surface of MXene and grow along a crystal orientation to form nanorod aggregates[41]. And then the aggregates are crosslinked as well as self-assembled into tubular structures, which might be attributed to the inductive effect between MXene and $CoC_2O_4$ to achieve low surface energy.

The scanning electron microscopy (SEM) images reveal that $CoC_2O_4$@MXene exhibits nanotube morphology in Fig. 1b, c. Notably, a rough surface is presented on $CoC_2O_4$@MXene compared with the pristine $CoC_2O_4$ nanorods (Supplementary Fig. 4). The special morphology is attributed to the MXene coating during the self-assembly of cobalt oxalate, while the ultrathin structure and surface functional groups of MXene nanosheets provide more flexibility for curling (Supplementary Figs. 5 and 6). Because of the opposite surface charges, the electrostatic adsorption force between MXene and $CoC_2O_4$ induces curling of MXene (Supplementary Fig. 7). As the crystal continues to grow, $CoC_2O_4$ evolves into a nanotube-like structure accompanied by an adhesive MXene layer (Fig. 1d, e), which is further demonstrated by the open-end hollow nanorods with a well-defined outline and certain curvature in Supplementary Fig. 8. The corresponding high-resolution transmission electron microscopy (HRTEM) image presents the typical boundary area of $CoC_2O_4$ and MXene with clearly different diffractive contrast in Fig. 1f, indicating that MXene is successfully coated on the surface of $CoC_2O_4$. Consistently, the lattice fringe with an interplanar spacing of 0.306 nm is indexed to the (114) plane of $CoC_2O_4$ (Fig. 1g), while a d-spacing of around 0.95 nm corresponds to the typical interlayer of MXene. Furthermore, the fast Fourier transformation image reveals the existence of the crystal phase of $CoC_2O_4$@MXene, which is consistent with the XRD results (Fig. 1h). Meanwhile, the uniform distribution mapping images of Co, Ti, O, and C elements also demonstrate the successful synthesis of MXene coated $CoC_2O_4$ (Fig. 1i). The tubular hybrid not only presents a higher specific surface area (Supplementary Fig. 9) but also exhibits abundant diffusion channels to activate the diffusion dimension of the electrolyte and unfreeze the restriction of mass transport. Even the electrolyte could enter the interior of the $CoC_2O_4$@MXene matrix, providing a larger electrolyte-electrocatalyst interface. Moreover, the introduction of highly hydrophilic MXene significantly improves water adsorption and electrode wettability, as shown by contact angle (CA) measurements (Supplementary Fig. 10). This unique structure provides the feasibility for spontaneous rapid complete reconfiguration.

### Electrochemical active site modulation

To correlate the rapid reconfiguration of $CoC_2O_4$@MXene with the improvement of catalytic activity during HER, the continuous LSV curves (5 mV $s^{-1}$ scan rate) were tested in 1 M KOH. The overpotential of $CoC_2O_4$@MXene decreased drastically with increasing LSV scan number to achieve a stable polarization curve after 5 cycles, indicating the possibility of reconfiguration of $CoC_2O_4$@MXene during HER (Fig. 2a). The corresponding overpotentials (@10 mA $cm^{-2}$) are plotted as a function of the LSV scan number in Fig. 2b. When the initial $CoC_2O_4$@MXene was subjected to continuous LSV scans for only 5 cycles, the overpotential strikingly decreases by approximately 46 mV, resulting in a low overpotential of 28 mV at 10 mA $cm^{-2}$. Moreover, chronopotentiometry measurement also further verified the reconfiguration phenomenon of $CoC_2O_4$@MXene (Supplementary Fig. 11).

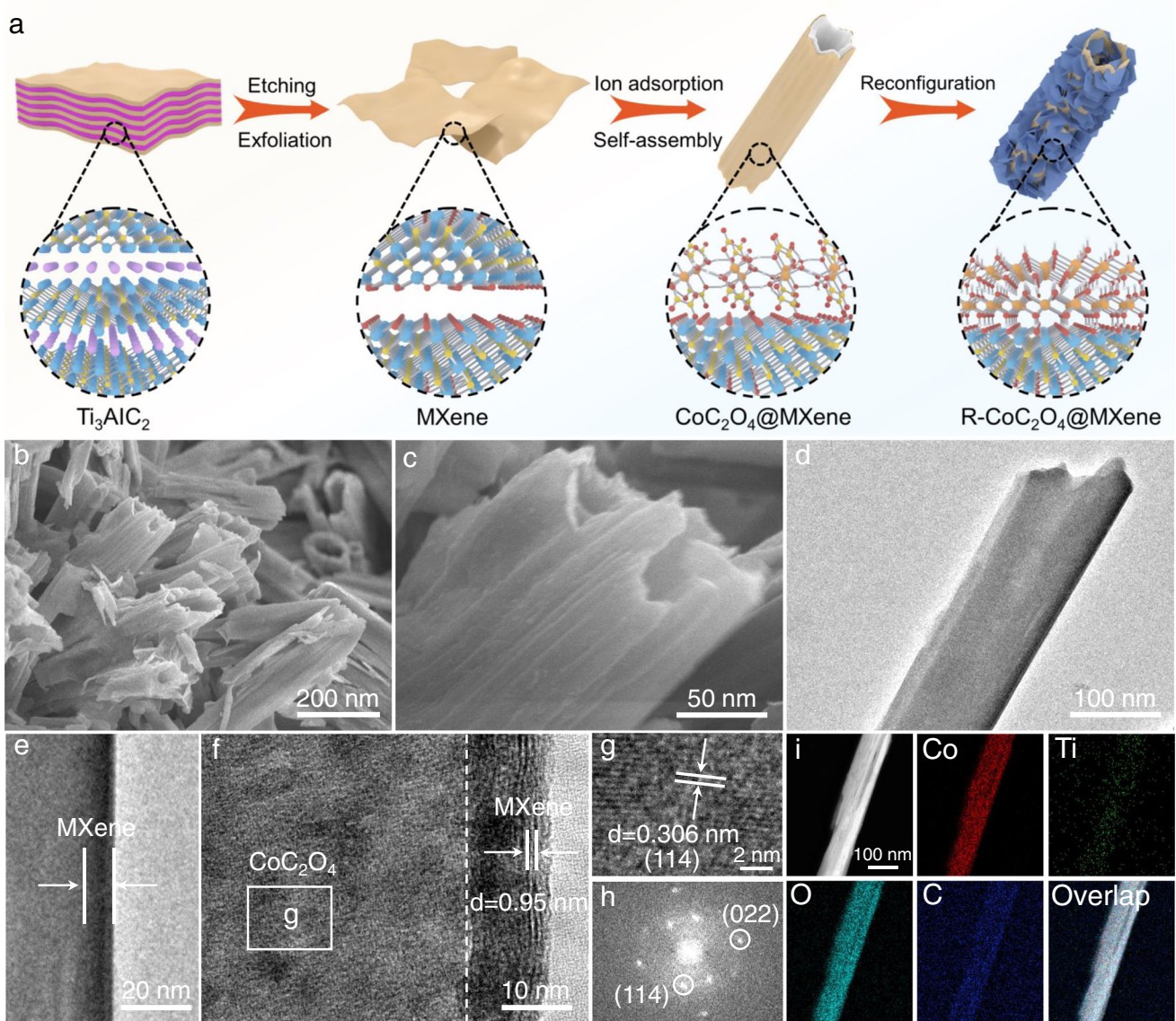

**Fig. 1 | Structural characterization of the fabricated CoC$_2$O$_4$@MXene precatalyst. a** Schematic illustration for the synthesis process of CoC$_2$O$_4$@MXene. **b**, **c** SEM, (**d**, **e**) TEM, (**f**, **g**) HRTEM images of CoC$_2$O$_4$@MXene. **h** Fast Fourier transformation (FFT) image. **i** Elemental mapping revealing the presence and homogenous distribution of Co, Ti, O, and C elements in CoC$_2$O$_4$@MXene.

CoC$_2$O$_4$@MXene exhibits the advantages of rapid reconfiguration and superior performance compared to recently reported HER precatalysts (Supplementary Table 1). The CoC$_2$O$_4$ without MXene coating also implies similar reconfiguration behavior, whereas the reconfiguration speed is tardy and the performance increase is much worse than CoC$_2$O$_4$@MXene (Supplementary Fig. 12). MXene can effectively enhance adsorbate accumulation and charge transfer through the analysis of mass transfer kinetics on the catalytic surface, which significantly facilitates the reconfiguration process (Supplementary Fig. 13). CoC$_2$O$_4$@MXene and CoC$_2$O$_4$ after electrochemical reconfiguration (repeated LSV scans) are denoted as R-CoC$_2$O$_4$@MXene and R-CoC$_2$O$_4$. These results demonstrate that the CoC$_2$O$_4$@MXene tubular catalyst with good hydrophilicity can increase the penetration of the electrolyte compared to the rod-shaped CoC$_2$O$_4$, benefiting the rapid reconfiguration.

The physical structure variations of the R-CoC$_2$O$_4$@MXene catalyst are firstly investigated to further explore the actual active species of HER after reconfiguration. Only peaks corresponding to Co(OH)$_2$ can be observed in the XRD patterns of R-CoC$_2$O$_4$@MXene, whereas R-CoC$_2$O$_4$ manifests an incompletely reconfigured multi-component composite structure (Supplementary Fig. 14). The morphological difference in catalysts before and after electrochemical treatment can be traced back to this transformation. R-CoC$_2$O$_4$@MXene exhibits a large number of hexagonal nanosheets interconnected to form a hollow rod-like loose structure (Supplementary Fig. 15), which allows deep penetration of alkaline electrolyte to stimulate catalytic reactions and complete reconfiguration. The TEM image of R-CoC$_2$O$_4$@MXene also shows the reconfigured loose nanorod structure and the ultrathin MXene nanosheets ideally adhere to nanorods (Supplementary Fig. 16). The HRTEM image reveals that these hexagonal nanosheets can be attributed to the formation of Co(OH)$_2$, in which the lattice fringes are 0.276 nm to comply with the (100) plane of Co(OH)$_2$ (Fig. 2c). In essence, the loose derivatization layer promotes the deep penetration of electrolyte, triggering a chain of evolutions till complete reconfiguration. However, the rod-like structure of pristine CoC$_2$O$_4$ is difficult to be completely hydrolyzed by the electrolyte, since the formed tight layer hinders the solution penetration for further reconfiguration (Supplementary Fig. 17). To reveal the special

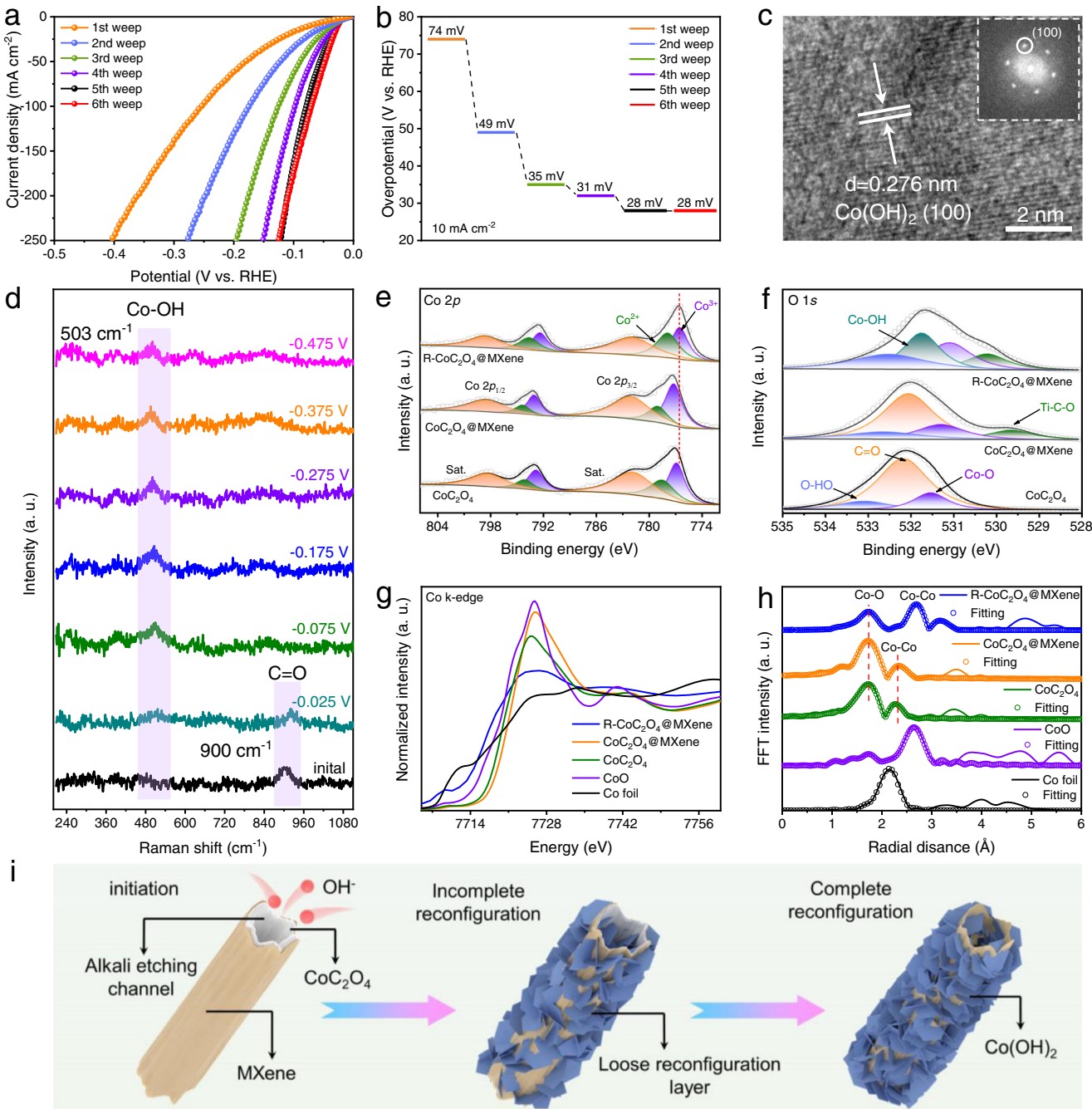

**Fig. 2 | Mechanism analysis of reconfiguration processes during HER.**
**a** Polarization curves of different LSV scans (IR correction) and (**b**) corresponding overpotentials of CoC$_2$O$_4$@MXene in 1 M KOH solution. **c** HRTEM image of R-CoC$_2$O$_4$@MXene. **d** In-situ Raman spectra of CoC$_2$O$_4$@MXene. XPS spectra of the (**e**) Co 2$p$ and (**f**) O 1$s$ regions for R-CoC$_2$O$_4$@MXene, CoC$_2$O$_4$@MXene, and CoC$_2$O$_4$ respectively. **g** Co K-edge XANES spectra and (**h**) FT-EXAFS spectra of the R-CoC$_2$O$_4$@MXene, CoC$_2$O$_4$@MXene, CoC$_2$O$_4$, CoO, and Co foil. **i** Reconfiguration process schematic diagram from CoC$_2$O$_4$@MXene to R-CoC$_2$O$_4$@MXene.

properties of reconfigured R-CoC$_2$O$_4$@MXene, Co(OH)$_2$@MXene is prepared by a classical method for comparison (Supplementary Figs. 18 and 19). The Co(OH)$_2$ in R-CoC$_2$O$_4$@MXene and the as-prepared Co(OH)$_2$@MXene have consistent crystal structures analyzed by SAED and HRTEM (Supplementary Fig. 20). Interestingly, the HRTEM of R-CoC$_2$O$_4$@MXene exhibits partial lattice ambiguity regions compared to Co(OH)$_2$@MXene, which implies the generation of defect structures. Moreover, R-CoC$_2$O$_4$@MXene inherits the excellent wettability of CoC$_2$O$_4$@MXene and outperforms the as-prepared Co(OH)$_2$@MXene. In-situ Raman spectra were applied to detect the reconfiguration process of the catalyst in real-time. Firstly, the local

structure of the initial CoC$_2$O$_4$@MXene is detected without applied potential and electrolyte, in which the detectable bands centered at 900 cm$^{-1}$ belong to the C=O vibration in CoC$_2$O$_4$ (Fig. 2d and Supplementary Fig. 21). Subsequently, the C=O vibration completely disappears when the potential was applied in 1 M KOH solution, while new broadband of 503 cm$^{-1}$ appears due to the Co−OH vibration in Co(OH)$_2$[24,25]. This is also consistent with the Co−OH characteristic peak positions in the as-prepared Co(OH)$_2$@MXene (Supplementary Fig. 22). This fact suggests that the application of potential in alkaline solution can trigger the rapid destruction of the C=O bond. Meanwhile, a large number of hydroxide ions in the electrolyte are immediately

supplied to form Co−OH coordination, which reveals the transformation mechanism of $CoC_2O_4$ to $Co(OH)_2$. In this regard, the reconfiguration process of the pre-catalyst $CoC_2O_4$@MXene is illustrated in Fig. 2i. XRD of $CoC_2O_4$@MXene immersed in 1 M KOH and performed neutral HER confirm the importance of the applied potential and alkali etching to drive the reconfiguration (Supplementary Fig. 23). The morphological difference between R-$CoC_2O_4$@MXene and $CoC_2O_4$@MXene soaked in 1 M KOH for 24 h intuitively indicates the driving of the reconfiguration by the applied potential (Supplementary Fig. 24).

The surface chemical states and electronic interaction during the reconfiguration process are examined by X-ray photoelectron spectroscopy (XPS) (Supplementary Fig. 25). The Co 2$p$ spectrum of $CoC_2O_4$@MXene is significantly shifted to higher binding energy (Fig. 2e), indicating that the valence state of the Co species is elevated. The higher valence Co atoms in $CoC_2O_4$@MXene favor electron acquisition, which provides a prerequisite for the reconfiguration of the catalyst. The Co 2$p$ of the reconfigured R-$CoC_2O_4$@MXene is shifted towards lower binding energy, indicating that the reconfiguration can further optimize the electronic structure of the catalyst. Moreover, The binding energy of the Co 2$p$ spectrum of R-$CoC_2O_4$@MXene is lower than that of $Co(OH)_2$@MXene due to the reduction potential of HER (Supplementary Fig. 26). Meanwhile, the Co 2$p$ spectrum of R-$CoC_2O_4$@MXene is negatively shifted compared with $CoC_2O_4$@MXene soaked in 1 M KOH for 24 h, which further verified the formation of low-valent Co species under HER potential conditions (Supplementary Fig. 27). Remarkably, the exfoliated MXene possesses a large number of C-Ti-O polar bonds, which facilitates the directional charge transfer between the host material and the carrier (Fig. 2f)[42]. Moreover, the introduction of MXene can alleviate the limitation of electron transmission and a continuous supply of electrons for the deep reconfiguration of $CoC_2O_4$. The appearance of Co−OH bonds in the O 1$s$ XPS spectra of R-$CoC_2O_4$@MXene also confirms the formation of $Co(OH)_2$.

X-ray absorption fine structure (XAFS) analysis is beneficial for the further investigation of the electronic structure and coordination environment of elements. The X-ray absorption near edge structure (XANES) of Co K-edge spectra of $CoC_2O_4$@MXene has a slight shift to higher energies compared to $CoC_2O_4$ (Fig. 2g), indicating the decreased electron density at the Co sites. When $Co^{2+}$ is oxidized to a high valence state, a charge transport orbit can be acquired for efficient electron transfer at the electrode-electrolyte interface, which provides a steady stream of electrons for catalyst reconfiguration. Additionally, the absorption edge of the R-$CoC_2O_4$@MXene shifts to lower energy relative to that of the $CoC_2O_4$@MXene, implying that the catalyst reconfiguration promotes charge redistribution, resulting in faster charge transfer in electrochemical reactions and enhanced catalytic activity. A similar oscillation with significant intensity reduction is observed for R-$CoC_2O_4$@MXene, indicating that the reconfiguration induces a different coordination environment (Supplementary Fig. 28). From extended XAFS (EXAFS) of Co K-edge (Fig. 2h), the dominant peak is assigned to the single scattering paths of the nearest Co−O shell, followed by one specific Co−Co shell. The fitting result of R-$CoC_2O_4$@MXene shows that the Co−Co distance in the first shell is 3.097 Å larger than that of $CoC_2O_4$ (2.652 Å) and $CoC_2O_4$@MXene (2.658 Å) (Supplementary Table 2), indicating the R-$CoC_2O_4$@MXene catalyst undergoes the reconfiguration[43]. The coordination number of Co−O decreases from 5.9 ($CoC_2O_4$@MXene) to 5.5 (R-$CoC_2O_4$@MXene), efficiently demonstrating the presence of coordinatively unsaturated Co sites[44]. These results are also confirmed in the wavelet transform (WT) (Supplementary Fig. 29). As shown in Supplementary Fig. 30, the adsorption edge of R-$CoC_2O_4$@MXene is shifted to lower energy compared to $Co(OH)_2$@MXene and $Co(OH)_2$, indicating the formation of low-valence Co species. The similar peak positions of Co−Co and Co−O bonds demonstrate the existence of $Co(OH)_2$ in

R-$CoC_2O_4$@MXene. More importantly, the electron paramagnetic resonance (EPR) detection found that R-$CoC_2O_4$@MXene possesses more oxygen defects than the $Co(OH)_2$@MXene prepared by the classical method (Supplementary Fig. 31), which provides the activity for the H adsorption site. This is also verified by the weaker Co−O bond in $CoC_2O_4$@MXene compared to $Co(OH)_2$@MXene in Co K-edge EXAFS spectra. The XPS and XANES analysis synergistically verify that R-$CoC_2O_4$@MXene induced by electron availability exhibits optimized electronic structure and exposed metal active sites, which lays the foundation for optimizing the intrinsic activity and increasing catalytic active sites.

## HER activity of reconfigured components

The electrocatalytic HER performance of various catalysts was examined in 1.0 M KOH to display the advantage of the in-situ reconfigured R-$CoC_2O_4$@MXene. The HER performance of R-$CoC_2O_4$@MXene synthesized with 20 mL MXene solution is the best among the counterparts with different ratios of $CoC_2O_4$ and MXene (Supplementary Fig. 32). The optimized R-$CoC_2O_4$@MXene requires a lower overpotential of 28 mV to achieve a current density of 10 mA cm$^{-2}$ compared to R-$CoC_2O_4$ (148 mV) and MXene (169 mV) (Fig. 3a). In particular, the current density of R-$CoC_2O_4$@MXene catalyst can reach 500 and 1000 mA cm$^{-2}$ at overpotentials of 157 and 216 mV, which are much higher than that of commercial Pt/C, and its highest current density can reach up to 1300 mA cm$^{-2}$ for potential industrial application. The corresponding Tafel slope of R-$CoC_2O_4$@MXene (43 mV dec$^{-1}$) is close to Pt/C (38 mV dec$^{-1}$), indicating strong HER kinetics (Fig. 3b)[45]. Electrochemical impedance spectroscopy (EIS) shows that R-$CoC_2O_4$@MXene has lower charge transfer resistance (Supplementary Fig. 33 and Table 3), which is attributed to the efficient electron transfer channel between MXene and low-valence $Co(OH)_2$. The high exchange current density also verifies the optimized electron transport (Supplementary Fig. 34). In addition, R-$CoC_2O_4$@MXene exhibits higher mass activity than commercial Pt/C (Supplementary Fig. 35). The higher turnover frequency (TOF) of R-$CoC_2O_4$@MXene verifies the excellent intrinsic activity (Fig. 3c). The electrochemical surface area (ECSA) value of R-$CoC_2O_4$@MXene is about 5.36 times higher than that of R-$CoC_2O_4$, revealing more HER active sites (Supplementary Fig. 36 and Table 4).

To understand the performance difference between R-$CoC_2O_4$@MXene and Pt/C catalysts, the release manner of the $H_2$ bubbles during HER was investigated to evaluate the mass transfer ability. The diameters of $H_2$ bubbles are small on R-$CoC_2O_4$@MXene (Fig. 3d), while they become larger with increasing current density on the Pt/C electrode. For R-$CoC_2O_4$@MXene, the majority of the detached bubbles were within the size of 0.2-0.3 mm at the current density of 1000 mA cm$^{-2}$, which is much smaller than the bubble diameter of the Pt/C electrode (Fig. 3e). These results indicate the superior mass transfer ability of R-$CoC_2O_4$@MXene at high current densities. The relationship between current density and Δη/Δlog |j| was analyzed to evaluate the high current density HER performance of R-$CoC_2O_4$@MXene (Fig. 3f)[35,46]. When the current density increases to 500 mA cm$^{-2}$, the ratio for R-$CoC_2O_4$@MXene (145 mV dec$^{-1}$) is much smaller than that Pt/C (220 mV dec$^{-1}$), but is still less than 174 mV dec$^{-1}$ for R-$CoC_2O_4$@MXene at a current density of 1000 mA cm$^{-2}$, indicating brilliant catalytic performance for high current density HER. A chronoamperometry test in Fig. 3g demonstrates that R-$CoC_2O_4$@MXene could maintain stability with little decay at current densities of 10, 500, and 1000 mA cm$^{-2}$ up to 100 h. The excellent durability is further confirmed by CV after 3000 cycles (Supplementary Fig. 37). After the durability test, R-$CoC_2O_4$@MXene still maintains its original phase, demonstrating the robust structure of R-$CoC_2O_4$@MXene (Supplementary Fig. 38). It is summarized that the HER performance of the R-$CoC_2O_4$@MXene is superior to most reported electrocatalysts in terms of both the low current density of 10 and high current densities

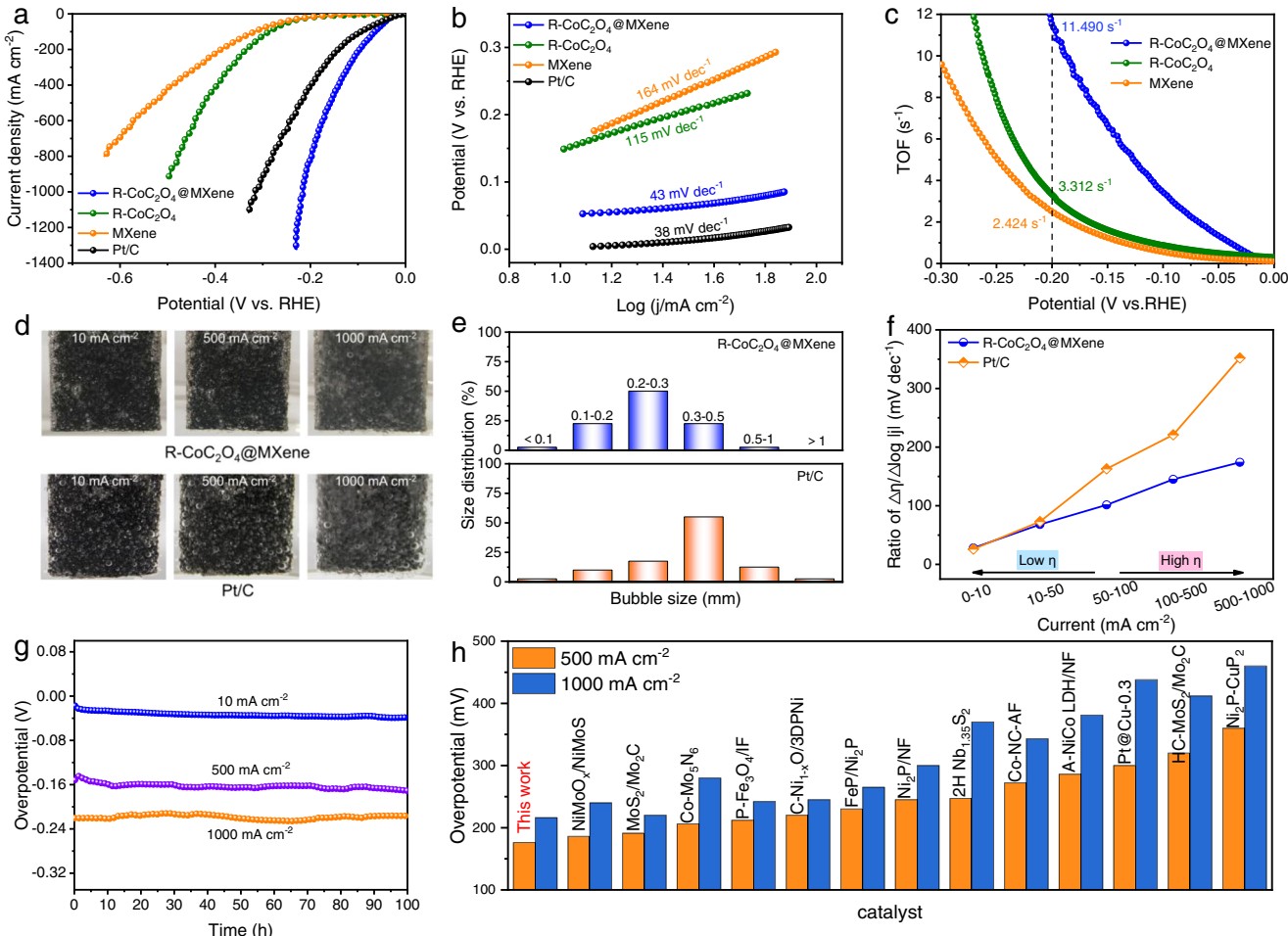

**Fig. 3 | Electrochemical HER performance measurements. a** The HER polarization curves and (**b**) Tafel slopes of MXene, R-CoC$_2$O$_4$, R-CoC$_2$O$_4$@MXene, and Pt/C. **c** The relationship between TOF and the tested potentials for MXene, R-CoC$_2$O$_4$, and R-CoC$_2$O$_4$@MXene. **d** Digital photos of H$_2$ bubbles during the HER process and (**e**) bubble size distributions on R-CoC$_2$O$_4$@MXene and Pt/C. **f** The ratios of $\Delta\eta$/$\Delta$log |j| for Pt/C and R-CoC$_2$O$_4$@MXene at different current densities. **g** Chronopotentiometry curves of R-CoC$_2$O$_4$@MXene at constant current densities of 10, 500, and 1000 mA cm$^{-2}$. **h** Comparison of overpotential (500 and 1000 mA cm$^{-2}$) of R-CoC$_2$O$_4$@MXene with recently reported high current density HER catalysts in 1 M KOH solution.

of 500 and 1000 mA cm$^{-2}$ (Fig. 3h and Supplementary Table 5). In addition, the catalytic activity and stability of R-CoC$_2$O$_4$@MXene are obviously superior to as-prepared Co(OH)$_2$@MXene in 1 M KOH (Supplementary Fig. 39). As a result, the reconfigured components deliver outstanding intrinsic activity and stability for HER.

## Theoretical calculation

To elucidate the relationship between the electronic structure of R-CoC$_2$O$_4$@MXene and the excellent HER catalytic activity, density functional theory (DFT) calculations were performed on the model structure (Supplementary Fig. 40). R-CoC$_2$O$_4$@MXene exhibits superior formation energies and relatively low antibonding states, verifying the stability of the computational model (Supplementary Figs. 41 and 42). The total density of states (DOS) displays that R-CoC$_2$O$_4$@MXene has a higher electron density near the Fermi level than CoC$_2$O$_4$@MXene and CoC$_2$O$_4$ (Fig. 4a and Supplementary Fig. 43). This result suggests that the complete reconfiguration of the catalyst enhances the electrical conductivity, which ensures fast electron transfer during electrocatalysis. Meanwhile, the d-band center shifts downward relative to the Fermi level from −0.514 eV in CoC$_2$O$_4$@MXene to −1.474 eV in R-CoC$_2$O$_4$@MXene (Fig. 4b)[47]. The d-band center downshift weakens the binding strength with H* and facilitates the desorption of H*, which optimizes the HER reaction process. From Fig. 4c, the differential charge density of

R-CoC$_2$O$_4$@MXene exhibits significant charge accumulation around the Co atom as well as dispersion of electron states around the O atom compared to CoC$_2$O$_4$ and CoC$_2$O$_4$@MXene (Supplementary Fig. 44). This electron redistribution behavior can optimize the reaction intermediates absorption energy, thus enhancing the catalytic activity[48].

The origins for HER catalytic activity improvement were further investigated using the intermediate free energy on the CoC$_2$O$_4$, CoC$_2$O$_4$@MXene, and R-CoC$_2$O$_4$@MXene systems. As shown in Fig. 4d, R-CoC$_2$O$_4$@MXene shows more negative adsorption energy of water molecule ($\Delta E_{H_2O}$) (−1.02 eV) compared to CoC$_2$O$_4$@MXene (−0.75 eV) and CoC$_2$O$_4$ (−0.41 eV) (Supplementary Fig. 45). This suggests that the exposed Co sites are beneficial to the adsorption and activation of H$_2$O. Subsequently, the activated H$_2$O molecules undergo dissociation, in which H and OH are adsorbed to the two adjacent unsaturated Co sites, respectively (Supplementary Fig. 46). The activation energy barrier for the R-CoC$_2$O$_4$@MXene system declines to 0.51 eV during the H$_2$O dissociation process, as lower than that of CoC$_2$O$_4$@MXene (0.76 eV) and CoC$_2$O$_4$ (1.04 eV) (Fig. 4e)[49]. The depressed Volmer step can be dramatically accelerated due to the positive effects on the thermodynamics of water adsorption and dissociation. Furthermore, the free energy of hydrogen adsorption ($\Delta G_{H*}$) is also an important factor to explain the HER activity, and the close to zero $\Delta G_{H*}$ value implies reasonable H adsorption and H$_2$ desorption. The $\Delta G_{H*}$ of R-CoC$_2$O$_4$@MXene is calculated to be 0.12 eV, which is

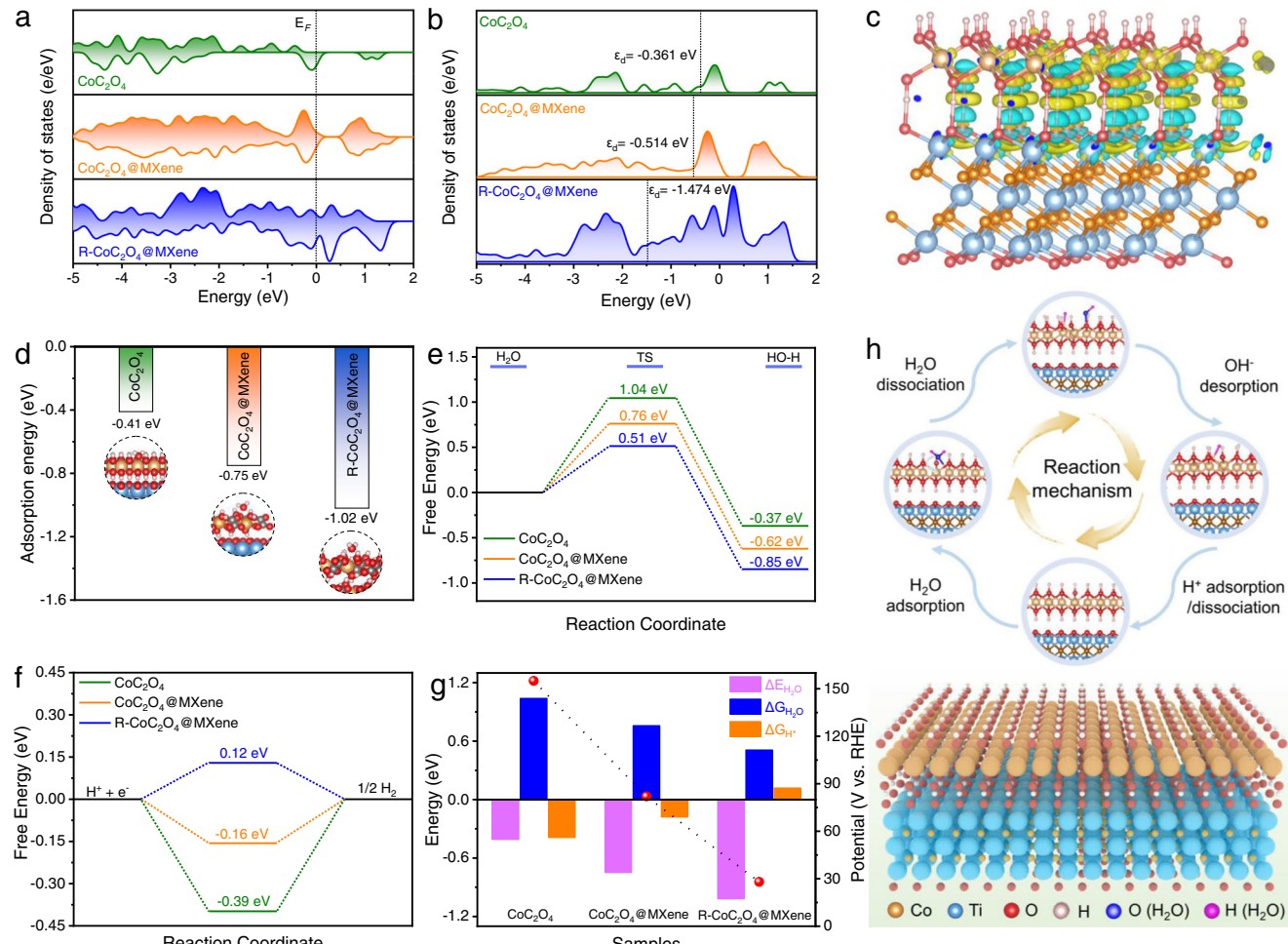

**Fig. 4 | Theoretical calculation of HER activation energy. a** The DOS plots and (**b**) D-band centers for $CoC_2O_4$, $CoC_2O_4$@MXene, and R-$CoC_2O_4$@MXene. **c** Differential charge density of R-$CoC_2O_4$@MXene, the yellow and blue zones represent the charge accumulation or dispersion, respectively. **d** $H_2O$ adsorption energy ($\Delta E_{H2O}$), (**e**) $H_2O$ dissociation energy ($\Delta G_{H2O}$), and (**f**) free energy of adsorbed H* intermediates ($\Delta G_{H*}$) on the $CoC_2O_4$, $CoC_2O_4$@MXene, and R-$CoC_2O_4$@MXene, respectively. **g** The relationship between the calculated $\Delta E_{H2O}$, $\Delta G_{H2O}$, $\Delta G_{H*}$ values, and the overpotential at 10 mA cm$^{-2}$. **h** Schematic illustration of the catalytic mechanism for alkaline HER on the R-$CoC_2O_4$@MXene.

closer to the thermoneutral value than that of the pristine $CoC_2O_4$ ($-0.39$ eV) and $CoC_2O_4$@MXene ($-0.16$ eV) (Fig. 4f). This indicates a suitable reaction energy barrier for H adsorption and $H_2$ desorption on unsaturated Co sites (Supplementary Fig. 47). Thus, the energy barrier parameters of different reaction stages in the HER process together determine the catalytic activity, leading to a reduction in the HER overpotential (Fig. 4g). Herein, the HER mechanism of exposed Co active sites in R-$CoC_2O_4$@MXene is revealed as shown in Fig. 4h. Overall, the R-$CoC_2O_4$@MXene with unsaturated coordination metal Co sites optimize the Gibbs free energy of HER pathways and electronic structure of the catalyst, so as to promote the HER process.

**Electrocatalytic seawater performance of the catalysts**

Seawater electrolysis is a promising alternative to conventional freshwater electrolysis due to its abundance and low cost, and it is necessary to evaluate the suitability of using seawater as an electrolyte in the electrolyzer[38]. Herein, the electrocatalytic HER activity of R-$CoC_2O_4$@MXene was measured in 1 M KOH seawater and natural seawater. R-$CoC_2O_4$@MXene exhibits the overpotentials of 32 and 163 mV at 10 mA cm$^{-1}$ in 1 M KOH seawater and natural seawater (Fig. 5a, b), respectively, which is considerably smaller than Pt/C (64 and 244 mV). Meanwhile, the corresponding Tafel slopes of the R-$CoC_2O_4$@MXene catalyst are calculated to be around 56 and 120 mV

dec$^{-1}$, reflecting the better HER kinetics in seawater (Fig. 5c). Moreover, according to the EIS results (Supplementary Fig. 48 and Table 6), a smaller semicircle is observed for R-$CoC_2O_4$@MXene relative to commercial Pt/C, inferring fast charge transfer in 1 M KOH seawater and neutral seawater. Remarkably, R-$CoC_2O_4$@MXene presents robust HER potential stability for 100 h in seawater (Fig. 5d). Meanwhile, compared with the as-prepared Co(OH)$_2$@MXene, R-$CoC_2O_4$@MXene also exhibits the best catalytic performance (Supplementary Fig. 49). After carefully evaluating those state-of-the-art catalysts previously reported, we list R-$CoC_2O_4$@MXene among the most efficient catalysts for HER in seawater (Fig. 5g and Supplementary Tables 7 and 8). Encouraged by the excellent HER performance of R-$CoC_2O_4$@MXene hybrids in seawater, an electrolytic cell was assembled to evaluate its potential for overall electrolysis (Fig. 5e). The R-$CoC_2O_4$@MXene can drive the overall seawater splitting with a cell voltage of 1.47 V (1.0 M KOH), 1.52 V (alkaline seawater), and 1.68 V (natural seawater) at 10 mA cm$^{-2}$, even outperforming the commercial Pt/C||RuO$_2$ (Fig. 5f and Supplementary Fig. 50). R-$CoC_2O_4$@MXene still exhibits outstanding durability after 30 h of continuous water electrolysis at a current density of 10 mA cm$^{-2}$ (Supplementary Fig. 51). Moreover, the water splitting performance of R-$CoC_2O_4$@MXene is superior to Co(OH)$_2$@MXene (Supplementary Fig. 52). The faraday efficiency (FE) was assessed by comparing the volumes of produced O$_2$ and H$_2$ during

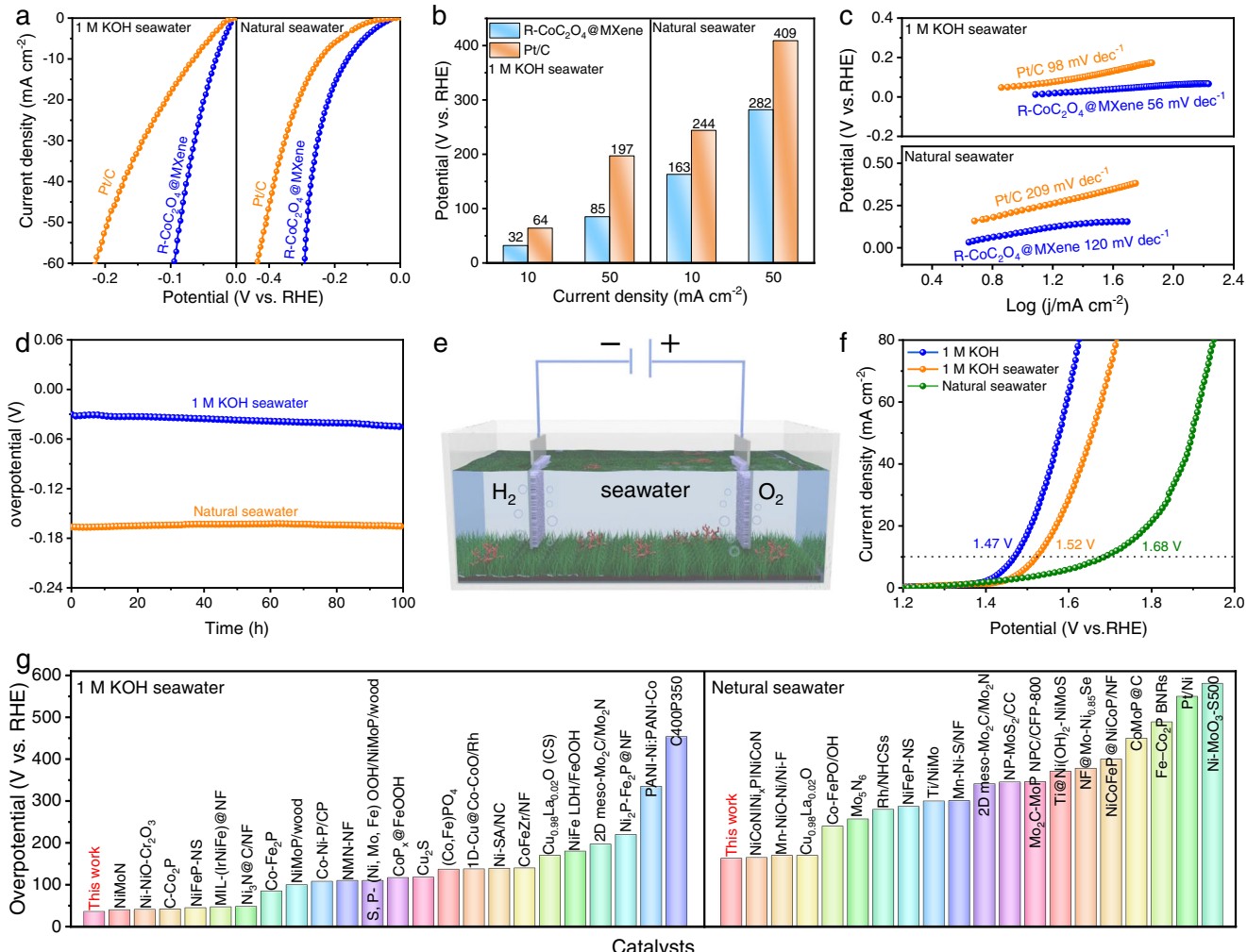

**Fig. 5 | Electrocatalytic seawater performance of the catalysts. a** the HER polarization curves, (**b**) overpotential comparison, and (**c**) Tafel slopes of R-CoC$_2$O$_4$@MXene and commercial Pt/C catalysts recorded in 1 M KOH seawater and natural seawater. **d** Chronopotentiometry curve of R-CoC$_2$O$_4$@MXene in 1 M KOH seawater and natural seawater. **e** Schematic illustration of the electrolyze. **f** Polarization curve of the R-CoC$_2$O$_4$@MXene ||R-CoC$_2$O$_4$@MXene toward overall water splitting in different electrolytes. **g** Comparison of HER overpotential (10 mA cm$^{-2}$) in 1 M KOH seawater and natural seawater.

the experiment process. The amount of evolved O$_2$ and H$_2$ matches well with the theoretical value in Supplementary Fig. 53, representing FE close to 100%. These results demonstrate that the R-CoC$_2$O$_4$@MXene catalyst has fully developed electrochemical performance, which makes it outstanding catalytic activity and stability in seawater electrolytes.

## Discussion

In summary, we have delicately designed a novel CoC$_2$O$_4$@MXene tubular catalyst with rapid complete reconfiguration properties to enhance HER performance. High electronic accessibility and abundant electrolyte diffusion channels induced the spontaneous and instantaneous reconfiguration of CoC$_2$O$_4$@MXene to form Co(OH)$_2$@MXene. Rapid reconfiguration can expose plenty of Co sites, thereby resulting in the creation of usable charge transfer orbitals and facilitating the electron transfer process. In-situ characterizations and DFT calculations reveal that the obtained Co(OH)$_2$@MXene after reconfiguration behave as the real active species, which can tune the electronic structure and optimize the energy barriers of water dissociation and H* intermediates, enhancing the intrinsic activity toward OER. As a result, the reconfigured CoC$_2$O$_4$@MXene electrocatalyst exhibits brilliant HER performance, which can acquire 10 and 1000 mA cm$^{-2}$ at a low overpotential of 28 and 216 mV in alkaline conditions. More

importantly, the HER activity of R-CoC$_2$O$_4$@MXene is stable in alkaline seawater and natural seawater, which is superior to commercial Pt/C. Our research provides an in-depth understanding into the in-situ reconfiguration of electrocatalysts during HER and facilitates the rational design and controllable synthesis of high-performance catalysts for sustainable hydrogen generation.

## Methods

### Materials

All the chemicals were used without further purification. Cobalt nitrate hexahydrate (Co(NO$_3$)$_2$·6H$_2$O), oxalic acid dihydrate (H$_2$C$_2$O$_4$·2H$_2$O), potassium hydroxide (KOH), Ti$_3$AlC$_2$ powders (200 mesh), lithium fluoride (LiF), hydrochloric acid (HCl), hexamethylenetetramine (HMT), and platinum on graphitized carbon (Pt/C, 20 wt%) were purchased from Macklin.

### Preparation of MXene nanosheets

1 g of Ti$_3$AlC$_2$ powders were slowly added to the mixed solution of 2 g of LiF and 30 ml of HCl and magnetically stirred in an oil bath at 45 °C for 36 h to selective removing of Al layers. Subsequently, the acidic mixed solution containing MXene was washed with distilled water several times by centrifugation (2264 xg) until the supernatant reached neutral pH (≥6). The collected precipitate was vacuum dried

to obtain bulk MXene powder. Finally, 0.2 g of MXene powder was dispersed in 300 ml of distilled water and sonicated under Ar atmosphere for 5 h, and then the supernatant was collected by centrifugation.

### Preparation of $CoC_2O_4$@MXene and $CoC_2O_4$

To prepare $CoC_2O_4$@MXene, 0.20 g of $H_2C_2O_4$ $2H_2O$ was dissolved in 20 mL of deionized water to obtain solution A, and then 0.46 g of $Co(NO_3)_2·6H_2O$ was dissolved in a mixed solution of 20 mL of MXene and 10 mL of deionized water to obtain solution B. Subsequently, the solution B was slowly added dropwise into solution A and stirred magnetically for 6 h at room temperature. Finally, the dispersion was collected by centrifugation at 7155 xg for 5 min and washed several times with deionized water, followed by freeze-drying for 12 h. For comparison, we performed control experiments with different (0, 10, 20, and 30 ml) MXene solution under other conditions unchanged, and the products can be labeled as $CoC_2O_4$, $CoC_2O_4$@MXene-10, $CoC_2O_4$@MXene-20 and $CoC_2O_4$@MXene-30, respectively.

### Preparation of $Co(OH)_2$@MXene and $Co(OH)_2$

0.36 g of $Co(NO_3)_2·6H_2O$ and 1.68 g of HMT were dissolved in a mixed solution of 150 mL of deionized water and 50 mL of ethanol. Then, 20 mL of MXene solution was added. Subsequently, the reaction solution was heated to about 90 °C for 1 h under magnetic stirring. Finally, the dispersion was collected by centrifugation at 7000 rpm for 5 min and washed several times with deionized water and ethanol, then vacuum dried for 10 h. MXene was not added during the preparation of $Co(OH)_2$.

### Characterizations

The structure and morphology were observed using scanning electron microscopy (SEM, Regulus 8100) and high-resolution transmission electron microscopy (HRTEM, FEI Tecnai G2 F20). The in-situ Raman spectra were collected by RENISHAW at an excitation wavelength of 633 nm. X-ray diffraction (XRD) data obtained from Bruker D8 Advance equipment was used to analyze the crystal structure. X-ray photoelectron spectrometer (XPS) with Al Kα X-rays was performed to study the surface composition of the samples. Bruker ENX-500 device was used to measure electron paramagnetic resonance (EPR) data. The Brunauer-Emmetand-Teller (BET) surface area was determined using the instrument V-Sorb 2008P. X-ray absorption fine structure spectra were measured under room temperature using the transmission mode of the XAFCA beamline in the Singapore Synchrotron Light Source. Extended X-ray absorption fine structure data were interpreted utilizing WINXAS 3.1 code, where it was normalized and then transformed to momentum space (k) from the initial energy space. The chemical states of the materials were studied by X-ray photoelectron spectroscopy (Thermo ESCALAB 250). The water contact angle was measured using a contact angle analyzer (DO4010 Easy drop, KRUSS), dropping 10 μL of water droplets from a height of 2.8 cm. The contact angle was recorded when the elapsed time after the water drop reached 1 min.

### Electrochemical measurement

Electrocatalytic activity tests were performed in a three-electrode system using an electrochemical workstation (Autolab Instrument). Graphite rod and a KCl-saturated Ag/AgCl electrode were used as the counter electrode (CE) and reference electrode (RE), respectively. The prepared catalyst, Super P, and binder (PVDF) were mixed in a mass ratio of 7:2:1 to acquire a catalyst slurry. The slurry was coated on $1 \times 1\,cm^2$ of nickel foam as the working electrode (loaded mass was around 1.5 mg cm$^{-2}$). The potentials were normalized to the reversible hydrogen electrode (RHE) by the Nernst equation ($E_{RHE} = E_{Ag/AgCl} + 0.198 + 0.0591 \times pH$)[50]. Polarization curves were measured by linear sweep voltammetry (LSV) at a scan rate of 5 mV s$^{-1}$ and compensated using iR. Cyclic

voltammograms (CV) were recorded at increasing scan rates (20–100 mV s$^{-1}$) within the Faradaic potential window (0.42–0.52 V vs. RHE) to obtain electrochemical surface area (ECSA). Electrochemical impedance spectroscopy (EIS) was performed in the frequency range of 0.1–105 Hz with an amplitude of 5 mV.

### Calculation methods

We have employed the first principles to perform all density functional theory (DFT) calculations within the generalized gradient approximation (GGA) and utilize the Perdew–Burke–sErnzerhof (PBE) formulation[51]. We have chosen projected enhanced wave (PAW) potentials to describe the ionic cores and take valence electrons into account using a plane-wave basis set with a kinetic energy cutoff of 450 eV. Partial occupancies of the Kohn-Sham orbitals were allowed by applying the Gaussian smearing method and a width of 0.05 eV. Convergence values for energy change and geometry optimization were set as 0.03 eV Å$^{-1}$ and $10^{-4}$ eV, respectively. Dispersive interactions were described by Grimme's DFT-D3 method. The Brillouin zone was sampled using a $1 \times 1 \times 1$ uniform Monkhorst-Pack k grid. Finally, the adsorption energies (Eads) were calculated as Eads = Ead/sub − Ead − Esub, where Ead/sub, Ead, and Esub are the total energies of the optimized adsorbate/substrate system, the adsorbate in the structure, and the clean substrate, respectively. The free energy was calculated using the equation:

$$G = E + APE - TS \qquad (1)$$

where G, E, ZPE, and TS are the free energy, total energy from DFT calculations, zero-point energy, and entropic contributions, respectively.

## Data availability

The data that support the findings of this study are available from the corresponding authors upon reasonable request. Source data are provided with this paper.

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

## Acknowledgements

This work was supported by the National Natural Science Foundation of China (51871119, S.-P., 22075141, S.-P., and 22101132, F.-H.), Scientific and Technological Innovation Special Fund for Carbon Peak and Carbon Neutrality of Jiangsu Province (BK20220039), Jiangsu Provincial Founds for Natural Science Foundation (BK20210311, S.-P.), China Postdoctoral Science Foundation (2021M691561, F.-H. and 2021T140319, F.-H.), Postgraduate Research & Practice Innovation Program of NUAA (xcxjh20210607, L.-W), Scientific and Technological Innovation Special Fund for Carbon Peak and Carbon Neutrality of Jiangsu Province (BK20220039, S.-P.), and Jiangsu Postdoctoral Research Fund (2021K547C, F.-H.).

## Author contributions

S.P. and L.W. conceived the project and designed the experiments. L.W. performed the synthesis and characterization of catalysts and the electrocatalytic measurements. Y.H., L.D., L.W., and S.Z. was involved in the structural and electrochemical analysis. L.W. wrote the paper. S.P., F.H., and L.L. contributed to revising the manuscript. All authors discussed the results and commented on the paper.

## Competing interests

The authors declare no competing interests.
