## [Peer Review File · Nature Communications]

Title: Rapid complete reconfiguration induced actual active species for industrial hydrogen evolution reactionREVIEWER COMMENTS

Reviewer #1 (Remarks to the Author):

In this study, Wang and co-workers have studied the HER activities of $\text{Co(OH)}_2/\text{MXenes}$ transformed electrochemical from $\text{CoC}_2\text{O}_4@\text{MXene}$ nanotube structures in alkaline media. They present quite an impressive overpotential and activity in seawater. They have utilized a wide range of characterization techniques and performed DFT calculations to identify the origin of the superior HER performance. This study is interesting, however, I may not be able to recommend it for publication in its current form. The structure and reaction mechanism proposed by DFT is not very clear and its relevance to the experimental observations must be discussed in great detail. Moreover, the following points must be addressed.

-In the introduction, the authors emphasize the sluggish kinetics of HER that limit the overall efficiency of water electrolysis, however, OER is even more sluggish, therefore the authors must acknowledge that alkaline electrolysis technology suffers more from OER kinetics.

-In the introduction (line 41) the authors used the term “electrochemical reconfiguration”, which is somewhat related to the surface structural reconfiguration. What do the authors mean by “electrochemical reconfiguration”?

-The authors attribute the flexibility of MXene nanosheets to the thinness and surface functional groups. It is however not clear what types of functional groups are present on the surface and how they take part in curling has not been explained. What makes MXene “adhesive” and how it is related to hydrophilicity require better explanation.

-It appears that CoC_2O_4 is covering both sides of the MXene nanosheet, there are catalytically active layers both inside and outside of the nanotubes. However, the authors emphasize the enhanced electrolyte transport inside the tubes, which are several tens of nm wide openings. BET surface area is larger, however larger available surface area for HER needs to be justified by ECSA measurements performed on as-prepared samples.

-What is the potential range of repeated LSVs? According to Figure 2a, the highest cathodic potential is -0.35 V vs. RHE. However, the Raman signature of Co(OH)_2 formation appears at -1.0 V. The authors must present in situ-Raman spectroscopy results in the potential regime relevant to the structural transformation.

-The authors utilize the combination of XPS and Co K-edge XANES-EXAFS to keep track of electronic structure change in the course of structural reconfiguration. First, they attribute 0.35 eV binding energy shift of $\text{CoC}_2\text{O}_4@\text{MXene}$ as compared to CoC_2O_4 electron transfer from CoC_2O_4 to the MXene. Then, one would expect binding energy to shift in opposite direction in Ti 2p and C 1s states of MXene. They associate structural changes with steady stream of electrons etc. I suggest the authors refrain from using such ill-defined descriptions to interpret energy shifts since they practically assume that CoC_2O_4 and $\text{CoC}_2\text{O}_4@\text{MXene}$ are structurally and compositionally similar. Co(OH)_2 formation is not apparent in XANES data, the authors must provide reference data from Co(OH)_2 .

The authors assign a peak in O 1s spectra to oxygen vacancies. Such assignments are quite common in literature. However, it is counterintuitive to any O 1s feature to oxygen not present on the surface. Most reduced oxides are readily hydroxylate, and a higher binding energy feature appears in XPS.

The authors must provide the structures and water dissociation and hydrogen adsorption/desorption processes in great detail and discuss the relevance of the model considered. They only considered a single layer of Co(OH)_2 , however, such information has not been provided by the experimental results. It is difficult to understand if water is dissociated on fully OH terminated Co(OH)_2 overlayer or Co with unsaturated coordination (with OH removed?). If the latter is the case the authors must provide DOS of this structure with 1, 2, 3 etc undercoordinated Co and demonstrate the stabilities. It has been shown that DOS at the Fermi level increases and the center of the d-band drops. Is this finding in accordance with the XPS and XANES? In addition, the elementary steps of Volmer steps must be well presented. It is not easy to see the site where water dissociation takes place, OH and H adsorption sites must be well described. The structural details of the H_2 desorption process and energy barrier (Tafel step) would also be beneficial.

Minor comments:

Figure 2 and 3 in the SI: the Authors should label the crystal planes and confirm the phase purity of Ti_3AlC_2 . They should also comment on why the reflection from (002) plane of MXene is absent in $\text{Co}_2\text{O}_4@ \text{Ti}_3\text{AlC}_2$.

Line 110: SEM should be scanning electron microscopy not Scan electron microscopy.

The title of the last section has to be "Conclusion" rather than "Discussion"

Figure 24 in the SI: Atoms must be labeled.

Reviewer #2 (Remarks to the Author):

In this work, the authors prepared heterostructures of Co_2O_4 coated by MXene nanosheets ($\text{Co}_2\text{O}_4@ \text{MXene}$) and observed the reconfiguration to the actual catalytic active species, Co(OH)_2 . Low overpotentials could be achieved with reasonable activity and stability. The work is technically sound, but several issues must be addressed before consideration for publication.

1. "Hydrogen fuel is the most promising alternative to fossil energy". Why and how so?
2. The cost for large-scale application is a complicated issue. Life cycle cost analysis is required. It is not just the "catalyst" itself. Rewording/rephrasing of authors' statement may be necessary.
3. The transformation of Co_2O_4 to Co(OH)_2 is spontaneous in alkaline solutions in OER, but how the rational design of catalysts is conceived for the reconfiguration under HER?
4. When authors described high, fast, rapid, deep, etc, there was no reference point. They should quantify these activities and compare these values to the state of the art systems.
5. The pre-edge feature of $\text{R-Co}_2\text{O}_4@ \text{MXene}$ looks different from others. What does it imply?
6. The DFT suggests that the TS energy barrier for $\text{R-Co}_2\text{O}_4@ \text{MXene}$ is -1.53eV, the highest among the three models!
7. How does the discovery teach the readers to better "design" HER catalysts? In other words, how general is this "strategy"?

Reviewer #3 (Remarks to the Author):

The manuscripts "Rapid complete reconfiguration induced actual active species for industrial hydrogen evolution reaction" by Luqi Wang, Yixin Hao, Liming Deng, Feng Hu, Sheng Zhao, Linlin Li, Shengjie Peng investigated the outstanding performance of HER from 1M KOH to neutral seawater. This excellent activity is attributed to the surface reconfiguration from CoC₂O₄@MXene to Co(OH)₂ by 1M KOH and applied potentials. MXene improves wettability and induced complete reconfiguration and converted Co(OH)₂ hexagonal plate provides unsaturated Co as active sites and such claim is supported by the in-situ Raman, X-ray spectroscopy, electrochemical tests and computational investigation. Despite the noticeable performances, the correlation between experimental results and fundamental explanation is yet to be enough for publication in Nature Communications. The following are suggestions for the authors to enhance the manuscript.

- 1) The superior HER activity of R-CoC₂O₄@MXene is driven from Co(OH)₂@MXene configuration by in-situ electrochemical activation. XRD identified R-CoC₂O₄@MXene as Co(OH)₂@MXene, therefore, the word "surface" reconfiguration is not appropriate but crystal phase changes. To emphasize surface reconfiguration by in-situ electrochemical activation compared to normally synthesized Co(OH)₂@MXene, all the data from characterization to catalytic activity and durability need to compare with Co(OH)₂@MXene. As-prepared Co(OH)₂@MXene is characterized only by XRD data (in Supplementary Figure 17). The authors should provide scanning electron micrography, transmission electron micrography with electron diffraction patterns, contact angles of the water droplets, Raman spectroscopy, X-ray photoelectron spectroscopy, X-ray absorption spectroscopy, electrochemical measurements in the medium from 1M KOH to neutral seawater. Please clarify different surfaces and their catalytic activity between reconfiguration by electrochemical methods vs. synthesized Co(OH)₂@MXene.
- 2) In addition to the contact angle of the water on Co(OH)₂@MXene, please provide those on MXene, R-CoC₂O₄@MXene, also on key catalysts after the electrochemical tests to consolidate wettability by MXene and their sustainability after catalysis.
- 3) In Figure 2a, surface reconfiguration of CoC₂O₄@MXene occurs during 5 cycles and CoC₂O₄ without MXene slowly changed over 20 cycles. The authors claimed MXene facilitates surface reconfiguration, but the underlying principle has not been suggested or elucidated. Please give more discussion on how MXene induces reconfiguration under the potentials and 1M KOH medium.
- 4) To clarify reconfiguration by electrochemical potentials and KOH medium, the authors should show whether reconfiguration of CoC₂O₄@MXene occurs or not when it is stored in KOH for long time without applying potentials and when applying potentials in a neutral medium.
- 5) In Supplementary Figure 11, please define crystal growth direction and exposed planes of Co(OH)₂@MXene assisted by electron diffraction patterns and d-spacing. Then compare with those of pristine MXene and as-prepared Co(OH)₂@MXene in the Supplementary Figure 17)
- 6) It would be good that the authors provide more EIS data with an insert for smaller ohm regions and a detail discussion on charge/mass transfers matching with materials characterization.
- 7) Minor revisions such as Line 110 scan electron microscopy (SEM), Supplementary Figure 13 without y-

axis(Potential) scale, Supplementary Figure 14 with missing indications in R-CoC2O4@MXene (e.g., lower than 400eV), and representing CV after 3000 cycles (text line 266) to Supplementary Figure 23 (polarization curves)

Point-by-Point Response to Review Comments

We sincerely appreciate the pertinent suggestions of editors and reviewers. The quality of the manuscript has been significantly improved following reviewers' comments. All modifications have been highlighted with yellow shading in the revised manuscript.

Reviewers' comments:

Reviewer #1: In this study, Wang and co-workers have studied the HER activities of $\text{Co}(\text{OH})_2/\text{MXenes}$ transformed electrochemical from $\text{CoC}_2\text{O}_4/\text{MXene}$ nanotube structures in alkaline media. They present quite an impressive overpotential and activity in seawater. They have utilized a wide range of characterization techniques and performed DFT calculations to identify the origin of the superior HER performance. This study is interesting. However, I may not be able to recommend it for publication in its current form. The structure and reaction mechanism proposed by DFT is not very clear and its relevance to the experimental observations must be discussed in great detail. Moreover, the following points must be addressed.

1 In the introduction, the authors emphasize the sluggish kinetics of HER that limit the overall efficiency of water electrolysis, however, OER is even more sluggish, therefore the authors must acknowledge that alkaline electrolysis technology suffers more from OER kinetics.

Our response: We thank the reviewer for the good suggestions. As pointed out by the reviewer, OER is more sluggish in the electrolyzed water system compared to HER. However, HER is also one of the important factors affecting the efficiency of water electrolysis. Therefore, we have revised the introduction section of the manuscript, according to this valuable suggestion. Please see Line 1-3, Page 2, highlighted in yellow.

2 In the introduction (line 41) the authors used the term “electrochemical reconfiguration”, which is somewhat related the surface structural reconfiguration. What do the authors mean by “electrochemical reconfiguration”?

Our response: We thank the reviewer for this valuable suggestion. In recent years, it has been noticed that catalysts exhibited a gradual improvement phenomenon during electrochemical testing [Angew. Chem. Int. Ed. 2020, 59, 3544; Chem. 2017, 3, 122-133.]. Extensive experimental studies have shown that the improved catalytic performance was attributed to the electrocatalysts undergoing structural reconfiguration or phase evolution during electrochemical oxidation or reduction, which was commonly defined as electrochemical reconfiguration. Meanwhile, most electrochemical reconfiguration only involves oxidation or reduction of the catalyst surface to generate oxides/(oxy)hydroxides or low-valent metal compounds, respectively, while the internal components remain unchanged [Angew. Chem. Int. Ed. 2020, 59, 21106; Nat Commun. 2022, 13, 605.]. This compositional change induced by oxidation or reduction of the catalyst surface is believed to undergo a surface reconfiguration. In addition, the crystal structures of the catalysts are completely transformed during the electrochemical process, which is defined as complete reconfiguration [Adv. Sci. 2022, 9, 2103567; Adv. Mater. 2020, 32, 2001136.]. As a result, surface reconfiguration is one type of electrochemical reconfigurations.

3 The authors attribute the flexibility of MXene nanosheets to the thinness and surface functional groups. It is however not clear what types of functional groups are present on the surface and how they take part in curling has not been explained. What makes MXene “adhesive” and how it is related to hydrophilicity require better explanation.

Our response: We thank the reviewer for this helpful comment. To further reveal the surface functional groups of MXene, FTIR spectroscopy was performed. As shown in Supplementary Fig. 5, the absorption bands at 568.28, 1110.63, and 1417.35 cm^{-1} correspond to the stretching vibrations of the -O, -F, and -OH functional groups on the MXene surface, respectively [Adv. Sci. 2019, 6, 1801470; Adv. Funct. Mater. 2020, 30, 1910028.]. Due to the presence of surface functional groups such as -O, -F, and -OH,

the MXene nanosheets are endowed with a negative charge, which is also demonstrated by the zeta potential of -23.34 mV (Supplementary Fig. 7). In contrast, CoC_2O_4 displays a zeta potential of +10.37 mV, indicating a positive surface charge. This opposite surface charge property enables the strong electrostatic interaction between MXene and CoC_2O_4 , which provides the possibility for MXene to curl and adhere to the surface of CoC_2O_4 [Adv. Funct. Mater. 2020, 30, 2002595; Carbon. 2021, 183, 872-883.]. Meanwhile, the TEM images of MXene exhibit ultrathin nanosheet structures with well-defined contours and transparent features in Fig. R1. Ultrathin 2D nanosheets endow MXene with unique flexibility and structural plasticity [Adv. Mater. 2020, 32, 2000919; Adv. Mater. 2017, 29, 1702367.]. In addition, the self-assembly of CoC_2O_4 with MXene is realized in the liquid phase, which is influenced by the fluid effect. The hydrophilic MXene can avoid the influence of fluid interaction and sufficient contact with CoC_2O_4 to trigger the electrostatic effect (Supplementary Fig. 10a), which favors the binding of MXene to CoC_2O_4 . [Nano Energy. 2019, 63, 103880; Adv. Sci. 2021, 8, 2101664.]. As a result, the ultrathin MXene nanosheets with structural plasticity can be curled and attached to the surface of CoC_2O_4 under the action of electrostatic adsorption force. The corresponding analysis was added in the revised manuscript and Supporting Information. Please see Line 29, Page 4, Line 1, Page 5, Supplementary Fig. 5, and Supplementary Fig. 7.

Supplementary Fig. 5. a) FTIR spectra of MXene, CoC_2O_4 , and CoC_2O_4 @MXene respectively. b) Enlarged FTIR spectrum of MXene.

Supplementary Fig. 7. Zeta spectra of MXene, CoC₂O₄, and CoC₂O₄@MXene, respectively.

Fig. R1. a) Low-magnified and b) high-magnified TEM images of MXene, respectively.

Supplementary Fig. 10. a) Contact angles of water droplets on MXene, CoC₂O₄, and CoC₂O₄@MXene, respectively.

4 It appears that CoC₂O₄ is covering both sides of the MXene nanosheet, there are catalytically active layers both inside and outside of the nanotubes. However, the authors emphasize the enhanced electrolyte transport inside the tubes, which are several tens of nm wide openings. BET surface area is larger, however larger available surface area for HER needs to be justified by ECSA measurements performed on as-prepared samples.

Our response: We thank the reviewer for the valuable suggestion. Electrochemical surface area (ECSA) is an important measure to determine the number of active sites.

$$ECSA = C_{dl}/C_s$$

The electrochemical surface area was measured by the capacitive method. As shown in Supplementary Fig. 33, the transient period during the anodic CV scans is the same and a clear overlap of both the anodic and cathodic steady current is observed. The electrochemical double capacitance (C_{dl}) in the Faradaic potential region is calculated by linear fitting. C_{dl} values of MXene, R-CoC₂O₄, and R-CoC₂O₄@MXene can be obtained from Supplementary Fig. 33d. Since the typical capacitance range is 20-60 $\mu\text{F cm}^{-2}$ per cm^2 , the capacitance of 40 $\mu\text{F cm}^{-2}$ per cm^2 is selected as standard capacitance to assess ECSA here. Thus, the ECSA is calculated to be 27.00, 29.00, and 155.50 cm^2 for MXene, R-CoC₂O₄, and R-CoC₂O₄@MXene respectively. Please see Supplementary Table 4 in the revised Supporting Information. The ECSA value of R-CoC₂O₄@MXene is approximately 5.36 times higher than that of R-CoC₂O₄, which is attributed to the presence of active layers both inside and outside the nanotubes, resulting in a large number of active sites. The corresponding analysis was added in the revised manuscript. Please see Line 18-20, Page 11, highlighted in yellow.

Supplementary Fig. 33. Cyclic voltammograms of a) R-CoC₂O₄@MXene, b) R-CoC₂O₄, and c) MXene at various scan rates (20, 40, 60, 80, and 100 mV s⁻¹) in the potential range of 0.42-0.52 V vs. RHE, which were used to estimate the double-layer capacitance (C_{dl}). d) Corresponding linear fitting of the current density versus scan rates.

Supplementary Table 4. The electrochemical active surface area of MXene, R-CoC₂O₄, and R-CoC₂O₄@MXene.

Sample	Slope value/mF cm ⁻²	ECSA/cm ²
MXene	1.08	27.00
R-CoC ₂ O ₄	1.16	29.00
R-CoC ₂ O ₄ @MXene	6.22	155.50

5 What is the potential range of repeated LSVs? According to Figure 2a, the highest cathodic potential is -0.35 V vs. RHE. However, the Raman signature of Co(OH)₂ formation appears at -1.0 V. The authors must present in situ-Raman spectroscopy

results in the potential regime relevant to the structural transformation.

Our response: We thank the reviewer for the significant comment. The potential range of the repeated LSVs is 0 ~ -0.5 V vs. RHE in the Fig. 2a. LSV has been corrected by IR in Fig. 2a. In Fig. 2d, the potential labels of the in situ Raman spectra are not converted corresponding to the RHE. Since the in situ Raman spectroscopy measurements were performed in 1 M KOH solution using Hg/HgO as the reference electrode, we have converted the potential relative to the RHE by the Nernst equation ($E_{\text{RHE}} = E_{\text{Hg/HgO}} + 0.098 + 0.0591 \times \text{pH}$) and changed the potential labels in Fig. 2d. In the revised Fig. 2d, it can be observed that the in situ Raman features of $\text{Co}(\text{OH})_2$ appear at -0.075 V vs. RHE, indicating that the structure of the $\text{CoC}_2\text{O}_4@\text{MXene}$ catalyst has been transformed at -0.075 V vs. RHE. We thank you again for your positive comments and valuable suggestions to improve the quality of our manuscript.

Fig. 2. a) Polarization curves of different LSV scans (IR correction) of $\text{CoC}_2\text{O}_4@\text{MXene}$ in 1 M KOH solution.

Fig. 2. d) In-situ Raman spectra of CoC₂O₄@MXene.

6 The authors utilize the combination of XPS and Co K-edge XANES-EXAFS to keep track of electronic structure change in the course of structural reconfiguration. First, they attribute 0.35 eV binding energy shift of CoC₂O₄@MXene as compared to CoC₂O₄ electron transfer from CoC₂O₄ to the MXene. Then, one would expect binding energy to shift in opposite direction in Ti 2p and C 1s states of MXene. They associate structural changes with steady stream of electrons etc. I suggest the authors refrain from using such ill-defined descriptions to interpret energy shifts since they practically assume that CoC₂O₄ and CoC₂O₄@MXene are structurally and compositionally similar. Co(OH)₂ formation is not apparent in XANES data, the authors must provide reference data from Co(OH)₂.

Our response: We thank the reviewer for the constructive comment to improve the quality of our work. Following the suggestion, we have carefully checked the description of energy transfer in the paper and made revisions. Please see Line 21-23, Page 8 in the revised manuscript, highlighted in yellow. In addition, XAFS reference data for Co(OH)₂ and Co(OH)₂@MXene has been added. Please see Supplementary Fig. 27 in the revised manuscript. The Co K-edge XANES spectra show that the adsorption edge of R-CoC₂O₄@MXene is close to Co(OH)₂. Meanwhile, from Co K-edge EXAFS spectra of Co(OH)₂, the dominant peak is assigned to the shell

coordination of Co-O and Co-Co, respectively. The dominant peak positions of Co-O and Co-Co shells in R-CoC₂O₄@MXene and Co(OH)₂ are almost the same, indicating that the geometrical structures of Co atoms are similar. These results further verify the formation of Co(OH)₂ in R-CoC₂O₄@MXene. The corresponding analysis was added in the revised manuscript. Please see Line 25-28, Page 9, highlighted in yellow.

Supplementary Fig. 27. a) Co K-edge XANES and b) FT-EXAFS spectra of the R-CoC₂O₄@MXene, Co(OH)₂, and Co(OH)₂@MXene, respectively.

7 The authors assign a peak in O 1s spectra to oxygen vacancies. Such assignments are quite common in literature. However, it is counterintuitive to any O 1s feature to oxygen not present on the surface. Most reduced oxides are readily hydroxylate, and a higher binding energy feature appears in XPS.

Our response: We thank the reviewer for the valuable comment. According to the suggestion, we have carefully examined the O 1s spectrum of R-CoC₂O₄@MXene and made reasonable corrections. In the revised Fig. 2f, the peak shape of Co-OH appears at the high binding energy of 531.75 eV, which further confirms the metal hydroxylation. The corresponding analysis was added in the revised manuscript. Please see Line 4-5, Page 9. In addition, oxygen vacancies have been demonstrated by EPR and EXAFS. As shown in Supplementary Fig. 28, the EPR signal can be found at a g value of 2.003. The stronger signal of R-CoC₂O₄@MXene indicates a higher level of O vacancies. The presence of oxygen vacancies is also confirmed by the weaker Co-O bonds in R-CoC₂O₄@MXene compared with Co(OH)₂@MXene in the Co K-edge

EXAFS spectra (Supplementary Fig. 27b).

Fig. 2. f) XPS spectra of O 1s regions for R-CoC₂O₄@MXene, CoC₂O₄@MXene, and CoC₂O₄ respectively.

Supplementary Fig. 28. EPR of R-CoC₂O₄@MXene and as-prepared Co(OH)₂@MXene, respectively.

Supplementary Fig. 27. a) Co K-edge XANES and b) FT-EXAFS spectra of the R-CoC₂O₄@MXene, Co(OH)₂, and Co(OH)₂@MXene, respectively.

8 The authors must provide the structures and water dissociation and hydrogen adsorption/desorption processes in great detail and discuss the relevance of the model considered. They only considered a single layer of Co(OH)₂, however, such information has not been provided by the experimental results. It is difficult to understand if water is dissociated on fully OH terminated Co(OH)₂ overlayer or Co with unsaturated coordination (with OH removed?). If the latter is the case the authors must provide DOS of this structure with 1, 2, 3 etc undercoordinated Co and demonstrate the stabilities. It has been shown that DOS at the Fermi level increases and the center of the d-band drops. Is this finding in accordance with the XPS and XANES? In addition, the elementary steps of Volmer steps must be well presented. It is not easy to see the site where water dissociation takes place, OH and H adsorption sites must be well described. The structural details of the H₂ desorption process and energy barrier (Tafel step) would also be beneficial.

Our response: We thank the reviewer for this helpful comment. Multilayer Co(OH)₂ was considered when constructing the computational model, but a single layer of Co(OH)₂ was ultimately chosen to more accurately reflect the interaction between Co(OH)₂ and MXene. As shown in Fig. 4c, the electron cloud density change of the R-CoC₂O₄@MXene heterostructure mainly occurs in the lower part of Co(OH)₂, which changes the charge properties of the upper part of Co sites. The optimized Co sites

facilitate the adsorption and desorption of reaction intermediates, resulting in an accelerated HER reaction rate. Further increasing the thickness of the Co(OH)_2 layer greatly weakens the interaction between MXene and Co(OH)_2 , which is not conducive to the analysis of catalyst performance changes. Similar theoretical calculations for heterostructured catalysts are also widely applied [Adv. Mater. 2018, 30, 1801171; Adv. Mater. 2020, 32, 2000872; Angew. Chem. Int. Ed. 2020, 59, 7245; Adv. Funct. Mater. 2021, 31, 2102117.].

The stronger signal fluctuation at g value of 2.003 and the weaker Co-O bond in the Co K-edge EXAFS spectrum confirm the existence of unsaturated coordinating Co in $\text{R-CoC}_2\text{O}_4@\text{MXene}$ (Supplementary Fig. 27 and 28). Therefore, unsaturated coordinated Co sites are rationally constructed in the structural model of $\text{R-CoC}_2\text{O}_4@\text{MXene}$ ($\text{R-CoC}_2\text{O}_4@\text{MXene}$ (1O_v)) (Supplementary Fig. 36). To further explore the effect of different numbers of unsaturated Co sites on the structural stability, we also construct $\text{R-CoC}_2\text{O}_4@\text{MXene}$ models with different oxygen vacancy contents ($\text{R-CoC}_2\text{O}_4@\text{MXene}$ (2O_v) and $\text{R-CoC}_2\text{O}_4@\text{MXene}$ (3O_v)) (Supplementary Fig. 37). Compared with $\text{R-CoC}_2\text{O}_4@\text{MXene}$ (2O_v) (-1.27 eV) and $\text{R-CoC}_2\text{O}_4@\text{MXene}$ (3O_v) (-0.32 eV), $\text{R-CoC}_2\text{O}_4@\text{MXene}$ (1O_v) (-1.98 eV) exhibits the lowest formation energy, indicating the most superior structural stability (Supplementary Fig. 38a). In addition, the total density of states display a shift towards higher energies with the increase in unsaturated Co sites (Supplementary Fig. 38b). The increase of antibonding orbitals leads to a decrease in the stability of the system. As a result, the $\text{R-CoC}_2\text{O}_4@\text{MXene}$ model with unsaturated Co sites ($\text{R-CoC}_2\text{O}_4@\text{MXene}$ (1O_v)) adopted in this manuscript exhibits the most superior stability, indicating the rationality of the theoretical calculations.

As shown in Fig. 4a and b, $\text{CoC}_2\text{O}_4@\text{MXene}$ has more antibonding empty orbitals than CoC_2O_4 , which is attributed to the elevated Co valence. These antibonding orbitals are favorable for accommodating more electrons to form excellent electron channels. It is worth noting that both the bonding and antibonding orbitals of $\text{R-CoC}_2\text{O}_4@\text{MXene}$ have more electron occupation, which makes the d-orbital electrons more delocalized, resulting in a lower d-band center. Therefore, the Co valence state of R-

$\text{CoC}_2\text{O}_4@\text{MXene}$ is lower than that of $\text{CoC}_2\text{O}_4@\text{MXene}$, which is also consistent with the results of XPS and XANES (Fig. 2e and g).

To enable a clearer understanding of the HER mechanism, we provide detailed computational structures and water dissociation/hydrogen adsorption processes. Please see Supplementary Fig. 36 and 41-43 in the revised Supporting Information, highlighted in yellow. As shown in Supplementary Fig. 41, H_2O is adsorbed on the unsaturated coordinating Co sites in $\text{R-CoC}_2\text{O}_4@\text{MXene}$. Meanwhile, $\text{R-CoC}_2\text{O}_4@\text{MXene}$ exhibits lower H_2O adsorption energy compared with the saturated Co sites in CoC_2O_4 and $\text{CoC}_2\text{O}_4@\text{MXene}$, indicating faster H_2O adsorption. Next, their structures for H_2O -dissociation are established (Supplementary Fig. 42). The activated H_2O molecules undergo dissociation, in which H and OH are adsorbed to the two adjacent unsaturated Co sites, respectively. Finally, ΔG_{H^*} is closer to the thermoneutral value on the unsaturated Co sites, implying reasonable H adsorption and H_2 desorption (Supplementary Fig. 43). The corresponding analysis has been added in the revised manuscript. Please see Line 3-5, Page 13, Line 22-23, Page 13, and Line 8-10, Page 14, highlighted in yellow.

Supplementary Fig. 27. a) Co K-edge XANES and b) FT-EXAFS spectra of the $\text{R-CoC}_2\text{O}_4@\text{MXene}$, Co(OH)_2 , and $\text{Co(OH)}_2@\text{MXene}$, respectively.

Supplementary Fig. 28. EPR of R-CoC₂O₄@MXene and as-prepared Co(OH)₂@MXene, respectively.

Supplementary Fig. 37. Top and side views of the structural models of R-CoC₂O₄@MXene with different oxygen defect contents.

Supplementary Fig. 38. a) Formation energy and b) DOS plots of the R-CoC₂O₄@MXene (10V), R-CoC₂O₄@MXene (20V), and R-CoC₂O₄@MXene (30V) structural models, respectively.

Fig. 4. a) The DOS plots and b) D-band centers for CoC₂O₄, CoC₂O₄@MXene, and R-CoC₂O₄@MXene.

Fig. 2. e) XPS spectra of the Co 2p for R-CoC₂O₄@MXene, CoC₂O₄@MXene, and CoC₂O₄, respectively. g) Co K-edge XANES spectra of the R-CoC₂O₄@MXene, CoC₂O₄@MXene, CoC₂O₄, CoO, and Co foil.

Supplementary Fig. 36. Top and side views of the structural models of a) CoC_2O_4 , b) $\text{CoC}_2\text{O}_4@MXene$, and c) $R\text{-CoC}_2\text{O}_4@MXene$, respectively.

Supplementary Fig. 41. Top and side views of the simulated H_2O adsorption structures of a) CoC_2O_4 , b) $\text{CoC}_2\text{O}_4@MXene$, and c) $R\text{-CoC}_2\text{O}_4@MXene$.

Supplementary Fig. 42. Top and side views of the simulated H₂O dissociation structures of a) CoC₂O₄, b) CoC₂O₄@MXene, and c) R-CoC₂O₄@MXene.

Supplementary Fig. 43. Top and side views of the simulated H adsorption structures of a) CoC₂O₄, b) CoC₂O₄@MXene, and c) R-CoC₂O₄@MXene.

9 Minor comments:

Figure 2 and 3 in the SI: the Authors should label the crystal planes and confirm the phase purity of Ti₃AlC₂. They should also comment on why the reflection from (002) plane of MXene is absent in CoC₂O₄@Ti₃AlC₂.

Line 110: SEM should be scanning electron microscopy not Scan electron microscopy.

The title of the last section has to be “Conclusion” rather than “Discussion”

Figure 24 in the SI: Atoms must be labeled.

Our response: We thank the reviewer for this helpful comment. Based on your suggestion, we have marked the corresponding crystal planes of Ti_3AlC_2 . In Supplementary Fig. 2, the characteristic peaks of Ti_3AlC_2 can correspond exactly to the crystal planes, which confirms the purity of Ti_3AlC_2 . The restacking of MXene nanosheets can be effectively suppressed, due to the MXene coating on the CoC_2O_4 surface. Therefore, the characteristic peak of the (002) plane interlayer spacing of MXene disappears in the $\text{CoC}_2\text{O}_4@\text{MXene}$ sample. “Scan electron microscopy” has been revised as “scanning electron microscopy”. “Discussion” has been revised as “Conclusion”. In addition, we have also added the corresponding atomic labels (Supplementary Fig. 36). We thank you again for your positive comments and valuable suggestions to improve the quality of our manuscript.

Supplementary Fig. 2. XRD pattern of Ti_3AlC_2 .

Supplementary Fig. 36. Top and side views of the structural models of a) CoC_2O_4 , b) $\text{CoC}_2\text{O}_4@\text{MXene}$, and c) $\text{R-CoC}_2\text{O}_4@\text{MXene}$, respectively.

Reviewer #2 (Remarks to the Author): In this work, the authors prepared heterostructures of CoC_2O_4 coated by MXene nanosheets ($\text{CoC}_2\text{O}_4@\text{MXene}$) and observed the reconfiguration to the actual catalytic active species, $\text{Co}(\text{OH})_2$. Low overpotentials could be achieved with reasonable activity and stability. The work is technically sound, but several issues must be addressed before consideration for publication.

1. “Hydrogen fuel is the most promising alternative to fossil energy”. Why and how so?

Our response: We thank the reviewer for the good comment. Fossil fuels based on coal and petroleum are rapidly depleting and their utilization is also causing the release of toxic and greenhouse gases into the atmosphere, leading to environmental pollution and global warming. Therefore, it is important to find renewable energy sources to replace fossil fuels. To date, solar, wind, bioethanol, biodiesel, hydrogen, and biogas have been used as alternative energy sources [Nat. Clim. Change. 2018, 8, 560-561; Adv. Energy Mater. 2018, 8, 1701620.]. Compared with other renewable energy sources, hydrogen fuel has a wide range of sources and various preparation methods, which can realize energy utilization across regions and seasons. Meanwhile, hydrogen fuel has a large storage capacity and can continuously release energy for a long time. In addition, the

product of hydrogen fuel is only water without any pollution to achieve complete zero carbon emissions [Renew. Sust. Energ. Rev. 2022, 167, 112698; Adv. Mater. 2020, 32, 1806326.]. It is difficult for other renewable energy sources to meet these advantages at the same time. Therefore, hydrogen fuel can be considered the most promising alternative to fossil energy.

2. The cost for large-scale application is a complicated issue. Life cycle cost analysis is required. It is not just the “catalyst” itself. Rewording/rephrasing of authors’ statement may be necessary.

Our response: We thank the reviewer for the valuable comment. Based on your suggestion, we have rewritten the introduction. The statement of "catalyst large-scale application" is also canceled. Please see the introduction section in the revised manuscript. We thank you again for your positive comments and valuable suggestions to improve the quality of our manuscript.

3. The transformation of CoC_2O_4 to $\text{Co}(\text{OH})_2$ is spontaneous in alkaline solutions in OER, but how the rational design of catalysts is conceived for the reconfiguration under HER?

Our response: We thank the reviewer for the good comment. Numerous reports have established that the intrinsic activity of the catalyst was derived from the reconfigured product for OER [Nat. Catal. 2019, 2, 763-772; Cell Rep. Phys. Sci. 2020, 1, 100241.]. The transition-metal-based catalysts have been regarded as stable for HER compared with harsh OER conditions. In recent years, it has been noticed that some catalysts may underwent structural reconfiguration or phase evolution during the reduction process [ACS Energy Lett, 2020, 5, 2483-2491; Angew. Chem. Int. Ed. 2020, 59, 3544.]. These catalysts reconfigured in the HER process have sparked great interest. We summarize the characteristics of HER catalyst reconfiguration to obtain the reasons for inducing this phenomenon. On the one hand, catalytic reaction conditions are important factors affecting catalyst reconfiguration, such as electrochemical operation, applied potential, electrolyte concentration, and pH. On the other hand, the intrinsic properties of HER

catalysts are also an important influencing factor because the reconfiguration is essentially a chemical reaction. According to numerous reports, partial dissolution, ion doping, and heterostructure are the main factors inducing the reconfiguration of HER catalysts [Adv. Energy Mater. 2022, 2201713; Adv. Mater. 2021, 33, 2007344.]. To achieve reconfiguration of HER catalysts, they can be designed with high surface area (favorable for partial dissolution), structural feasibility (eg, via ion doping), or efficient electron/ion transport properties (eg, via heterostructure construction). Herein, we obtain $\text{CoC}_2\text{O}_4@\text{MXene}$ heterostructure catalysts with nanotube-like structures through a self-assembly strategy. This favorable hydrophilic nanotubular structure can greatly increase the partial dissolution of the catalyst. Moreover, the ultra-thin MXene with high conductivity provides an effective electronic path, which facilitates the rapid and deep self-reconfiguration process. Therefore, designing pre-catalysts by combining a series of rational strategies is expected to achieve catalyst reconfiguration under HER conditions.

4. When authors described high, fast, rapid, deep, etc, there was no reference point. They should quantify these activities and compare these values to the state of the art systems.

Our response: We thank the reviewer for the significant comment. According to your suggestions, we have quantified the reconfiguration time, reconfiguration degree, and catalytic activity and compared them with state-of-the-art systems. Specifically, compared with previous HER reconfiguration catalysts, the $\text{CoC}_2\text{O}_4@\text{MXene}$ requires only 5 cycles (5 mV s^{-1} scan rate and potential range 0 ~ -0.5 V vs. RHE) of consecutive LSV scans (8.3 min) to achieve complete reconfiguration. The reconfigured $\text{CoC}_2\text{O}_4@\text{MXene}$ delivers small overpotentials of 28, 157, and 216 mV to reach current densities of 10, 500, and 1000 mA cm^{-2} in 1 M KOH, respectively. The advantages of the $\text{CoC}_2\text{O}_4@\text{MXene}$ catalyst are further demonstrated by quantifying the reconfiguration time, reconfiguration degree, and catalytic activity. Please see Supplementary Table 1 and 5 in the revised manuscript, highlighted in yellow.

Supplementary Table 1. Summary of reconfiguration product, reconfiguration degree,

reconfiguration time, and HER performance of different pre-catalysts in 1 M KOH.

Pre-catalyst	Reconfiguration product	Reconfiguration degree	Reconfiguration time	Overpotential (mV) 10 mA cm ⁻²	References
CoC₂O₄@MXene	Co(OH)₂@MXene	Complete reconfiguration	5 cycles LSV (8.3 min)	28	This Work
Ni ₄ Mo	Ni ₄ Mo@Mo ₂ O ₇ ²	Surface reconfiguration	4 h	30	Angew. Chem. Int. Ed. 2021, 60, 7051-7055.
NiSe ₂	NiSe/NiSe ₂	Deep reconfiguration	20 cycles LSV	50	ACS Energy Lett. 2020, 5, 2483-2491.
TiO ₂ Co ₂ P ₄ O ₁₂	CoO _x (OH) _y /TiO ₂ Co ₂ P ₄ O ₁₂	Surface reconfiguration	35 cycles LSV	50	J. Mater. Chem. A. 2019, 7, 12457-12467.
CoF ₂	Co(OH) ₂	Complete reconfiguration	60 cycles LSV	54	Adv. Sci. 2022, 9, 2103567.
Co _{3-x} Ni _x O ₄	Co _y Ni _{1-y} O/Co _{3-x} Ni _x O ₄	Surface reconfiguration	20 h	57	ACS Catal. 2021, 11, 8174-8182.
Mo ₂ C-Mo(VI)O _x /CC	Mo ₂ C-Mo(IV/VI)O _x /C	Surface reconfiguration	100 cycles LSV	60	Angew. Chem. Int. Ed. 2020, 59, 3544-3548.
TiO ₂ @CoCH	Co(OH) ₂ /TiO ₂ @CoCH	Surface reconfiguration	25 cycles LSV	66	Nano Energy. 2021, 82, 105732.
NF-NiS ₂	Ni/NF-NiS ₂	Surface reconfiguration	100 cycles LSV	67	Nano Energy. 2017, 41, 148-153.

Ni-BDT	NiNSs/ Ni-BDT	Surface reconfiguration	1000 cycles LSV	80	Chem. 2017, 3, 122-133.
CoFeO@BP	CoP/CoFeO@BP	Surface reconfiguration	20 cycles LSV	88	Angew. Chem. Int. Ed. 2020, 59, 21106-21113.
CoSe _{1.26} P _{1.42}	Co/CoSe _{1.26} P _{1.42}	Deep reconfiguration	N.A.	92	ACS Energy Lett. 2019, 4, 987-994.
CoP	Co(OH) _x /CoP	Surface reconfiguration	10 h	100	Chem. Sci. 2019, 10, 2019-2024.
Co _{0.5} W _{0.5} S _x	CoO/Co(OH) ₂ /C _{0.5} W _{0.5} S _x	Surface reconfiguration	10 h	110	J. Mater. Chem. A. 2021, 9, 11359-11369.
CoNP@C	Co(OH) ₂ /CoNP@C	Surface reconfiguration	3 h	191	Nano Energy. 2018, 49, 14-22.

Supplementary Table 5. Comparison of HER activity measured for R-CoC₂O₄@MXene with other representative reported HER catalysts using 1.0 M KOH as electrolyte.

Sample	Overpotential (mV) 10 mA cm ⁻²	Overpotential (mV) 500 mA cm ⁻²	Overpotential (mV) 1000 mA cm ⁻²	References
R-CoC₂O₄@MXene	28	157	216	This work
2H Nb _{1.35} S ₂	46	247	370	Nat Mater. 2019, 18, 1309-1314.
W ₁ Mo ₁ -NG DAC	67	N.A.	N.A.	Sci. Adv. 2020, 6, eaba6586.
NFP/C-3	95	N.A.	N.A.	Sci. Adv. 2019, 5, eaav6009.

Fe(OH) _x @Cu-MOF NBs	112	N.A.	N.A.	Sci. Adv. 2021, 7, eabg2580.
NiMoO _x /NiMoS	38	186	240	Nat. Commun. 2020, 11, 5462.
MoS ₂ /Mo ₂ C	45	191	220	Nat. Commun. 2019, 10, 269.
FeP/Ni ₂ P	85	230	265	Nat. Commun. 2018, 9, 2551.
CF/VMFP	45	230	N.A.	Nat. Commun. 2021, 12, 1380.
MoC-Mo ₂ C-790	98.2	292	N.A.	Nat. Commun. 2021, 12, 6776.
HC-MoS ₂ /Mo ₂ C	280	320	412	Nat Commun. 2020, 11, 3724.
PW-Co ₃ N NWA/NF	41	N.A.	N.A.	Nat. Commun. 2020, 11, 1853.
Ni-Fe NP	46	N.A.	N.A.	Nat. Commun. 2019, 10, 5599.
NiCo-SAD-NC	61	N.A.	N.A.	Nat Commun. 2021, 12, 6766.
3CoMo-Vs	75	N.A.	N.A.	Nat Commun. 2020, 11, 2253.
1T _{0.81} -MoS ₂ @NiS ₂	95	N.A.	N.A.	Nat Commun. 2021, 12, 5260.
MoC-Mo ₂ C-790	98.2	N.A.	N.A.	Nat Commun. 2021, 12, 6776.
LSC/K-MoSe ₂	128	N.A.	N.A.	Nat Commun. 2021, 12, 4606.
Co-doped CeO ₂	75	240	N.A.	J. Am. Chem. Soc. 2020, 142, 6461- 6466.

Ni ₂ P/NF	85	245	300	J. Am. Chem. Soc. 2019, 141, 7537- 7543.
Co-NC-AF	85	272	343	Adv.Mater. 2021, 33, 2103533.
P-Fe ₃ O ₄ /IF	39	212	242	Adv. Mater. 2019, 31, 1905107.
Cr _{0.4} Mo _{0.6} B ₂	180	390	N.A.	Adv. Mater. 2020, 32, 2000855.
CuCo-CAT/CC	52	N.A.	N.A.	Adv. Mater. 2021, 33, 2106781.
CoNi-inf	72	N.A.	N.A.	Adv. Mater. 2020, 32, 2002857.
Ni/Ni(OH) ₂	77	N.A.	N.A.	Adv. Mater. 2020, 32, 1906915.
Ni@N-HCGHF	95	N.A.	N.A.	Adv. Mater. 2020, 32, 2003313.
MoO ₂ -FeP@C	103	N.A.	N.A.	Adv. Mater. 2020, 32, 2000455.
Ni ₄ Mo	34	N.A.	N.A.	Angew. Chem. Int. Ed. 2021, 60, 7051- 7055.
Ni ₃ N-Co ₃ N PNAs/NF	43	N.A.	N.A.	Angew. Chem. Int. Ed. 2021, 60, 5984- 5993.
S-NiFe ₂ O ₄	61	N.A.	N.A.	Angew. Chem. Int. Ed. 2021, 60, 14117- 14123.
Co-MoS ₂ @CoS ₂	76	N.A.	N.A.	Angew. Chem. Int. Ed. 2022, 61, e2021148.
CoFeO@BP	88	N.A.	N.A.	Angew. Chem. Int. Ed. 2020, 59, 21106- 21113.

P-MoP/Mo ₂ N	89	N.A.	N.A.	Angew. Chem. Int. Ed. 2021, 60, 6673-6681.
CoP _x @CNS	91	N.A.	N.A.	Angew. Chem. Int. Ed. 2020, 59, 21360-21366.
N-c-CoSe ₂	98	N.A.	N.A.	Angew. Chem. Int. Ed. 2021, 60, 21575-21582.
S-Co _{0.85} Se-1	108	N.A.	N.A.	Angew. Chem. Int. Ed. 2021, 60, 12360-12365.
2D meso-Mo ₂ C/Mo ₂ N	110	N.A.	N.A.	Angew. Chem. Int. Ed. 2022, 134, e202112298.
NiP ₂ -650	134	N.A.	N.A.	Angew. Chem. Int. Ed. 2021, 60, 259-267.
Co-MoS ₂	137	N.A.	N.A.	Angew. Chem. Int. Ed. 2021, 60, 7251-7258.
Co _{0.9} Ni _{0.1} Se	185.7	N.A.	N.A.	Angew. Chem. Int. Ed. 2020, 59, 22743-22748.
Co-Mo ₅ N ₆	19	206	280	Adv. Energy Mater. 2020, 10, 2002176.
C-Ni _{1-x} O/3DPNi	32	220	245	Adv. Energy Mater. 2020, 10, 2002955.
Ni/FeOOH	38	287	N.A.	Adv. Energy Mater. 2020, 10, 1904020.
NFN-MOF/NF	87	293	N.A.	Adv. Energy Mater. 2018, 8, 1801065.
E-Co SAs	59	280	N.A.	Adv. Funct. Mater. 2021, 31, 2100547.

Ni-Mo-B HF	23	329	N.A.	Adv. Funct. Mater. 2021, 32, 2107308.
PS-Cu	121	750	980	Adv. Funct. Mater. 2022, 2112367.
NiMnOP/NF	81	195	N.A.	Nano Energy. 2020, 69, 104432.
NiFe-LDH/MXene/NF	132	205	N.A.	Nano Energy. 2019, 63, 103880.
MFN-MOFs(2:1)/NF	79	234	N.A.	Nano Energy. 2019, 57, 1-13.
Ni ₂ P-CuP ₂	51	360	460	ACS Nano. 2021, 15, 5586-5599.
Ni ₃ Sn ₂ S ₂ @Ni ₃ S ₂	50	206	N.A.	Appl. Catal. B: Environ. 2020, 267, 118675.
NiFeMo IOS@NF	33	249	N.A.	Appl. Catal. B: Environ. 2020, 267, 118376.
A-NiCo LDH/NF	43	286	381	Appl. Catal. B: Environ. 2020, 261, 118240.
IrFe/NC	37	510	850	Appl. Catal. B: Environ. 2019, 258, 117965.
NF@NiFeLDH	166	356	N.A.	Small. 2021, 18, 2104354.

5. The pre-edge feature of R-CoC₂O₄@MXene looks different from others. What does it imply?

Our response: We thank the reviewer for the good comment. As shown in Fig. 2g, compared with CoC₂O₄@MXene, the pre-edge feature of R-CoC₂O₄@MXene is shifted to lower energy, indicating that the valence state of Co atoms is further reduced. The formation of low-valence Co(OH)₂ is attributed to the reduction potential during

the HER process [Chem 2017, 3, 122-133; ACS Energy Lett. 2019, 4, 987-994.], which is one of the important reasons for the improved catalytic activity. To further analyze the difference in pre-edge features, we have also added XANES and EXAFS of Co(OH)_2 and $\text{Co(OH)}_2\text{@MXene}$. Please see Supplementary Fig. 27 in the revised manuscript. The XANES of the Co K-edge spectrum of $\text{R-CoC}_2\text{O}_4\text{@MXene}$ has a slight shift to lower energy compared to Co(OH)_2 and $\text{Co(OH)}_2\text{@MXene}$, indicating a lower electron density at Co sites. The Co-O and Co-Co shell peak positions of $\text{R-CoC}_2\text{O}_4\text{@MXene}$ and Co(OH)_2 are almost identical, further verifying the formation of Co(OH)_2 after HER reconfiguration. These results confirm that $\text{CoC}_2\text{O}_4\text{@MXene}$ generates low-valence Co(OH)_2 during the HER process at the reduction potential, resulting in a distinct difference in the pre-edge feature of $\text{R-CoC}_2\text{O}_4\text{@MXene}$.

Fig. 2. g) Co K-edge XANES spectra of the $\text{R-CoC}_2\text{O}_4\text{@MXene}$, $\text{CoC}_2\text{O}_4\text{@MXene}$, CoC_2O_4 , CoO , and Co foil .

Supplementary Fig. 27. a) Co K-edge XANES spectra and b) EXAFS spectra of the R-CoC₂O₄@MXene, Co(OH)₂, and Co(OH)₂@MXene, respectively.

6. The DFT suggests that the TS energy barrier for R-CoC₂O₄@MXene is -1.53 eV, the highest among the three models!

Our response: We thank the reviewer for the good comment. As a physical process, H₂O adsorption only has Van der Waals force without correction of free energy and entropy. However, the H₂O dissociation process is a chemical reaction, which requires the calculation of the change in free energy. Therefore, it is inappropriate to regard the energy barrier between the H₂O adsorption and H₂O dissociation processes as the transition state energy barrier. The catalytic reaction energy barrier diagram has been modified to better distinguish the energy barrier changes for H₂O adsorption and H₂O dissociation. As shown in Fig. 4d, R-CoC₂O₄@MXene (-1.02 eV) displays much negative adsorption energies than CoC₂O₄@MXene (-0.75 eV) and CoC₂O₄ (-0.41 eV), suggesting that the R-CoC₂O₄@MXene hybrid structure is more favorable for the adsorption and activation of H₂O and provides support for the subsequent water dissociation. Then, the activation energy barrier for the R-CoC₂O₄@MXene system is reduced to 0.51 eV during the H₂O dissociation process (Fig. 4e), which is lower than that of CoC₂O₄@MXene (0.76 eV) and CoC₂O₄ (1.04 eV), indicating that the H-OH bond is more effortlessly broken down on R-CoC₂O₄@MXene. Overall, R-CoC₂O₄@MXene possesses the most superior Gibbs free energy barrier, which facilitates the HER process.

Fig. 4. d) H₂O adsorption energy and e) H₂O dissociation energy on the CoC₂O₄, CoC₂O₄@MXene, and R-CoC₂O₄@MXene, respectively.

7. How does the discovery teach the readers to better “design” HER catalysts? In other words, how general is this “strategy”?

Our response: We thank the reviewer for the valuable comment. Current studies found that electrocatalysts underwent dynamic reconfiguration and were transformed into actual active species during the HER process, leading to improved catalytic performance [Angew. Chem. Int. Ed. 2020, 59, 21106; Nano Energy 2021, 82, 105732; ACS Energy Lett. 2020, 5, 2483-2491; Joule 2020, 4, 673-687.]. The reconfiguration degree of most HER catalysts is limited, which leads to reduced component utilization and prevents the full exploitation of catalytic activity. Considering these aspects, the rational design of catalysts with special structural and chemical properties to trigger complete reconfiguration is an effective strategy to prepare high-performance catalysts. Herein, we rationally design a novel CoC₂O₄@MXene electrocatalyst with rapid complete reconfiguration properties in alkaline electrolytes for enhanced HER performance. These special rapid complete reconfiguration properties can be attributed to the following points. CoC₂O₄@MXene with a hydrophilic nanotube-like structure greatly increases the penetration of the electrolyte, which provides a basis for catalyst reconfiguration. Partial dissolution of CoC₂O₄ occurs during the electrochemical process and a loose reconfiguration layer is formed, leading to the complete transformation of the catalyst. Moreover, the ultrathin MXene with high electrical conductivity provides an efficient electronic pathway for catalyst reconfiguration. These results demonstrate that the reconfiguration process can be effectively regulated by rationally designing the morphology, properties, and electronic structure of the catalyst. Therefore, tuning the reconfiguration process to generate abundant active sites with high intrinsic activity is an effective strategy to improve the catalytic performance of electrocatalysts, opening a new avenue for high-performance electrocatalysts.

Reviewer #3 (Remarks to the Author): The manuscripts “Rapid complete

reconfiguration induced actual active species for industrial hydrogen evolution reaction” by Luqi Wang, Yixin Hao, Liming Deng, Feng Hu, Sheng Zhao, Linlin Li, Shengjie Peng investigated the outstanding performance of HER from 1M KOH to neutral seawater. This excellent activity is attributed to the surface reconfiguration from $\text{CoC}_2\text{O}_4@\text{MXene}$ to $\text{Co}(\text{OH})_2$ by 1M KOH and applied potentials. MXene improves wettability and induced complete reconfiguration and converted $\text{Co}(\text{OH})_2$ hexagonal plate provides unsaturated Co as active sites and such claim is supported by the in-situ Raman, X-ray spectroscopy, electrochemical tests and computational investigation. Despite the noticeable performances, the correlation between experimental results and fundamental explanation is yet to be enough for publication in Nature Communications. The following are suggestions for the authors to enhance the manuscript.

1)The superior HER activity of $\text{R-CoC}_2\text{O}_4@\text{MXene}$ is driven from $\text{Co}(\text{OH})_2@\text{MXene}$ configuration by in-situ electrochemical activation. XRD identified $\text{R-CoC}_2\text{O}_4@\text{MXene}$ as $\text{Co}(\text{OH})_2@\text{MXene}$, therefore, the word “surface” reconfiguration is not appropriate but crystal phase changes. To emphasize surface reconfiguration by in-situ electrochemical activation compared to normally synthesized $\text{Co}(\text{OH})_2@\text{MXene}$, all the data from characterization to catalytic activity and durability need to compare with $\text{Co}(\text{OH})_2@\text{MXene}$. As-prepared $\text{Co}(\text{OH})_2@\text{MXene}$ is characterized only by XRD data (in Supplementary Figure 17). The authors should provide scanning electron micrography, transmission electron micrography with electron diffraction patterns, contact angles of the water droplets, Raman spectroscopy, X-ray photoelectron spectroscopy, X-ray absorption spectroscopy, electrochemical measurements in the medium from 1M KOH to neutral seawater. Please clarify different surfaces and their catalytic activity between reconfiguration by electrochemical methods vs. synthesized $\text{Co}(\text{OH})_2@\text{MXene}$.

Our response: We thank the reviewer for the good comment. HER or OER catalysts undergo surface structural reconfiguration while the internal composition remains unchanged during electrochemical testing, which is defined as surface reconfiguration. Herein, the complete reconfiguration of the crystal phase has been achieved by the

rational design of the catalyst. We refer to the crystal phase transition from $\text{CoC}_2\text{O}_4@\text{MXene}$ to $\text{Co(OH)}_2@\text{MXene}$ as complete reconfiguration. To demonstrate the superiority of complete reconfiguration, we only describe surface reconfiguration catalysts for comparison in the Introduction section. Therefore, we employ complete reconfiguration to represent the crystal structure transition in the paper, rather than the surface reconfiguration. According to your suggestions, we have supplemented the relevant characterization of the as-prepared $\text{Co(OH)}_2@\text{MXene}$ and performed a detailed comparison with $\text{R-CoC}_2\text{O}_4@\text{MXene}$. The SEM of $\text{Co(OH)}_2@\text{MXene}$ demonstrates that the regular hexagonal nanosheets are tightly connected with MXene, indicating that $\text{Co(OH)}_2@\text{MXene}$ is successfully prepared (Supplementary Fig. 18). The TEM images in Supplementary Fig. 19a further reveal that MXene with ultrathin structures adheres ideally to Co(OH)_2 nanosheets, which is consistent with the SEM results. The lattice fringes of 0.276 nm correspond to the (100) plane of Co(OH)_2 species (Supplementary Fig. 19b). This result indicates that the crystal growth directions of the Co(OH)_2 species in $\text{R-CoC}_2\text{O}_4@\text{MXene}$ and $\text{Co(OH)}_2@\text{MXene}$ are consistent. However, compared with $\text{Co(OH)}_2@\text{MXene}$, the lattice fringes of $\text{R-CoC}_2\text{O}_4@\text{MXene}$ are weaker and have many amorphous regions (Fig. 2c), which can be attributed to the formation of lattice defects [Adv. Energy Mater. 2021, 11, 2100141.]. These lattice defects can supply more catalytic active centers and boost catalytic performance [Adv. Mater. 2020, 32, 2006784.]. In addition, $\text{R-CoC}_2\text{O}_4@\text{MXene}$ inherits the superior wettability of $\text{CoC}_2\text{O}_4@\text{MXene}$ and is superior to $\text{Co(OH)}_2@\text{MXene}$ by contact angle measurement in Supplementary Fig. 10. The above results indicate that $\text{R-CoC}_2\text{O}_4@\text{MXene}$ not only has superior surface wettability but also contains a large number of defect structures, which is beneficial to the improvement of HER performance.

The characteristic peaks of the Raman spectrum of $\text{Co(OH)}_2@\text{MXene}$ at 454 and 507 cm^{-1} are attributed to the Co-O stretching mode (Supplementary Fig. 21), which is also consistent with the in situ Raman results of $\text{R-CoC}_2\text{O}_4@\text{MXene}$. To further explore the difference in the electronic structure and elemental coordination environment between $\text{R-CoC}_2\text{O}_4@\text{MXene}$ and $\text{Co(OH)}_2@\text{MXene}$, the surface composition of

Co(OH)₂@MXene was investigated by XPS and XAFS characterization. The Co 2p spectrum of Co(OH)₂@MXene in Supplementary Fig. 24b shows that the characteristic peaks of Co 2p_{3/2} and Co 2p_{1/2} can be observed at binding energies of 777.53 and 793.22 eV with two satellite peaks at 783.06 and 799.27 eV. Obviously, the binding energies of Co 2p_{3/2} and Co 2p_{1/2} for R-CoC₂O₄@MXene are lower than that of Co(OH)₂@MXene, indicating that the Co atoms in R-CoC₂O₄@MXene are reduced to lower valence states. Moreover, the O 1s spectrum of R-CoC₂O₄@MXene is shifted to lower binding energies compared to Co(OH)₂@MXene (Supplementary Fig. 24c). This peak shift may be caused by the difference in the Co-O coordination environment. As shown in Supplementary Fig. 27a, Co K-edge XANES spectra indicate that the adsorption edge of R-CoC₂O₄@MXene is slightly shifted to lower energies relative to Co(OH)₂@MXene and Co(OH)₂, further verifying the formation of low-valence Co species. From Co K-edge EXAFS spectra in R space (Supplementary Fig. 27b), the main peaks are assigned to Co-O and Co-Co shell coordination. In addition, the similar peak positions of Co-Co and Co-O bonds further demonstrate the existence of Co(OH)₂ in R-CoC₂O₄@MXene. Meanwhile, the coordination number of the Co-O bond in the R-CoC₂O₄@MXene is significantly lower than that in the Co(OH)₂@MXene, effectively indicating the presence of coordinatively unsaturated Co sites. The electrocatalytic performance of Co(OH)₂@MXene for HER was measured in 1 M KOH and seawater solution. As shown in Supplementary Fig. 35, 45, and 48, the catalytic performance and stability of R-CoC₂O₄@MXene are superior to Co(OH)₂@MXene in both 1 M KOH and seawater electrolytes. These results confirm that the R-CoC₂O₄@MXene obtained by electrochemical reconfiguration contains low-valence Co(OH)₂ with unsaturated coordination compared with as-prepared Co(OH)₂@MXene, which improves the electronic structure and coordination environment of the catalyst, resulting in superior HER activity. The corresponding analysis has been added to the revised manuscript. Please see Line 1-6, Page 8, Line 12-13, Page 8, Line 27-28, Page 8, Line 25-28, Page 9, Line 3-4, Page 10, Line 11-13, Page 12, Line 13-14, Page 15, and Line 1-2, Page 16, highlighted in yellow.

Supplementary Fig. 18. a) Low-magnified and b) high-magnified SEM images of as-prepared $\text{Co(OH)}_2@MXene$, respectively.

Supplementary Fig. 19. a) TEM and b, c) HRTEM images of as-prepared $\text{Co(OH)}_2@MXene$.

Fig. 2. c) HRTEM image of $R\text{-CoC}_2\text{O}_4@MXene$.

Supplementary Fig. 10. a-e) Contact angles of water droplets on MXene, CoC_2O_4 , Co(OH)_2 , $\text{Co(OH)}_2@\text{MXene}$, and $\text{CoC}_2\text{O}_4@\text{MXene}$, respectively. f-j) Contact angles of water droplets on MXene, CoC_2O_4 , Co(OH)_2 , $\text{Co(OH)}_2@\text{MXene}$, and $\text{CoC}_2\text{O}_4@\text{MXene}$ after HER, respectively.

Supplementary Fig. 21. Raman spectra of as-prepared $\text{Co(OH)}_2@\text{MXene}$.

Supplementary Fig. 24. a) full XPS survey spectrum of $\text{Co(OH)}_2@MXene$. XPS spectra of the b) Co 2p and c) O 1s regions for $\text{R-CoC}_2\text{O}_4@MXene$ and $\text{Co(OH)}_2@MXene$, respectively.

Supplementary Fig. 27. a) Co K-edge XANES spectra and b) EXAFS spectra of the $\text{R-CoC}_2\text{O}_4@MXene$, Co(OH)_2 , and $\text{Co(OH)}_2@MXene$.

Supplementary Fig. 35. a) The HER polarization curves of Pt/C, $\text{R-CoC}_2\text{O}_4@MXene$, and $\text{Co(OH)}_2@MXene$. b) Chronopotentiometry curves of $\text{Co(OH)}_2@MXene$ at constant current densities of 10 and 500 mA cm⁻².

Supplementary Fig. 45. a, b) the HER polarization curves of MXene, R-CoC₂O₄@MXene and Co(OH)₂@MXene catalysts in 1 M KOH seawater and natural seawater. c) Chronopotentiometry curve of Co(OH)₂@MXene in 1 M KOH seawater and natural seawater.

Supplementary Fig. 48. a) Polarization curve and b) long-term stability tests of the Co(OH)₂@MXene||Co(OH)₂@MXene toward overall water splitting in different electrolytes.

2) In addition to the contact angle of the water on Co(OH)₂@MXene, please provide those on MXene, R-CoC₂O₄@MXene, also on key catalysts after the electrochemical tests to consolidate wettability by MXene and their sustainability after catalysis.

Our response: We are grateful for the kind comments. According to your suggestions, we have supplemented the contact angle for Co(OH)₂@MXene, MXene, R-CoC₂O₄@MXene, and key catalysts after electrochemical testing. Please see Supplementary Fig. 10 in the revised Supporting Information, highlighted in yellow. The superior wettability of MXene can be seen from the contact angle test. Meanwhile,

the water adsorption and electrode wettability of the catalysts can be further improved in the presence of highly hydrophilic MXene. Besides, the reconfigured R-CoC₂O₄@MXene maintains good hydrophilicity, which is also superior to Co(OH)₂@MXene. The corresponding analysis was added in the revised manuscript. Please see Line 5-6, Page 8.

Supplementary Fig. 10. a-e) Contact angles of water droplets on MXene, CoC₂O₄, Co(OH)₂, Co(OH)₂@MXene and CoC₂O₄@MXene, respectively. f-j) Contact angles of water droplets on MXene, CoC₂O₄, Co(OH)₂, Co(OH)₂@MXene, and CoC₂O₄@MXene after HER, respectively.

3) In Figure 2a, surface reconfiguration of CoC₂O₄@MXene occurs during 5 cycles and CoC₂O₄ without MXene slowly changed over 20 cycles. The authors claimed MXene facilitates surface reconfiguration, but the underlying principle has not been suggested or elucidated. Please give more discussion on how MXene induces reconfiguration under the potentials and 1M KOH medium.

Our response: We thank the reviewer for the good comment. Herein, the effect of MXene on reconfiguration is attributed to the ultrathin structure, high electrical conductivity, and superior wettability. As shown in Fig. R1, the TEM image shows clear ultrathin MXene nanosheets with well-defined outlines and transparent features. The ultrathin structure is highly desirable to facilitate charge transfer and mass diffusion

during electrochemical reactions, providing a basis for catalyst reconfiguration. Studies have reported that metallic MXene exhibited excellent electrical conductivity [Adv. Mater. 2019, 31, 1903841; Adv. Energy Mater. 2021, 11, 2102388.]. The high conductivity provides a constant flow of electrons for the reconstruction process. Moreover, CoC_2O_4 is hydrophobic with a large CA of 117° , while MXene exhibits high hydrophilic properties (Supplementary Fig. 10). The water adsorption and wettability of $\text{CoC}_2\text{O}_4@\text{MXene}$ electrodes can be further improved in the presence of highly hydrophilic MXene. Meanwhile, R- $\text{CoC}_2\text{O}_4@\text{MXene}$ maintains superior wettability. This indicates that the catalyst keeps in close contact with the alkaline electrolyte during the reconfiguration process, which is beneficial to the transformation of the crystal phase structure.

To further understand the importance of MXene for reconfiguration, we investigate the mass/charge transport process. Since the reconfiguration process is essentially a chemical reaction, the rate is mainly controlled by charge transport and mass transfer. The mass and charge transfer processes are investigated by using a rotating disk electrode based on the thin film theory to explore the kinetics of the reconfiguration process. All test data are obtained on glassy carbon electrodes at 1200 rpm to exclude the effects of hydrodynamics and outer liquid diffusion dynamics. Therefore, the mass transfer resistance is only concentrated in the retention membrane layer. This suggests that the electrode process kinetics are controlled only by adsorption and chemical reactions. From the direct variation trend and size of the semicircle in the Nyquist plot (Supplementary Fig. 12), it can be found that the reaction kinetics and the hydroxide adsorption rates of CoC_2O_4 and $\text{CoC}_2\text{O}_4@\text{MXene}$ become faster with the increased cycle numbers. More importantly, the kinetic process of $\text{CoC}_2\text{O}_4@\text{MXene}$ is always better than that of CoC_2O_4 , indicating that more hydroxide adsorbate accumulation drives the reconfiguration reaction. However, CoC_2O_4 slows down the reconfiguration rate due to the large internal resistance and the formation of surface passivation film, which hinder charge transfer. Furthermore, $\text{CoC}_2\text{O}_4@\text{MXene}$ has a faster initial surface charge accumulation than CoC_2O_4 by Bode plot (Supplementary Fig. 12 c and d). The frequency peaks of $\text{CoC}_2\text{O}_4@\text{MXene}$ significantly decrease and

shift to higher frequencies with the increased cycle number, implying that the surface charge keeps increasing to induce complete reconfiguration. There is no significant transition in the Bode plot frequency of CoC_2O_4 after multiple cycles, which is difficult to cross the energy barrier of the reconfiguration reaction. In conclusion, we found that MXene could effectively enhance adsorbate accumulation and charge transfer by analyzing the kinetics of mass transfer on the catalytic surface, which provides a prerequisite for deep reconfiguration. The corresponding analysis was added in the revised manuscript. Please see Line 3-6, Page 7.

Fig. R1. a) Low-magnified and b) high-magnified TEM images of MXene, respectively.

Supplementary Fig. 10. Contact angles of water droplets on a) MXene, b) CoC_2O_4 , e) $\text{CoC}_2\text{O}_4@MXene$, and j) $\text{CoC}_2\text{O}_4@MXene$, respectively.

Supplementary Fig. 12. a, b) EIS and c, d) Bode plots of different sweep cycles for $\text{CoC}_2\text{O}_4@\text{MXene}$ and CoC_2O_4 , respectively.

4) To clarify reconfiguration by electrochemical potentials and KOH medium, the authors should show whether reconfiguration of $\text{CoC}_2\text{O}_4@\text{MXene}$ occurs or not when it is stored in KOH for long time without applying potentials and when applying potentials in a neutral medium.

Our response: We are grateful for the kind comments. We have added the XRD patterns of $\text{CoC}_2\text{O}_4@\text{MXene}$ after soaking in 1M KOH for 6, 12, and 24 hours, respectively. The relevant data has been added in the revised Supporting Information. Please see Supplementary Fig. 22. The XRD patterns of $\text{CoC}_2\text{O}_4@\text{MXene}$ after soaking in 1 M KOH for 6 h can be well matched with CoC_2O_4 . Meanwhile, $\text{CoC}_2\text{O}_4@\text{MXene}$ displays a composite structure of CoC_2O_4 and $\text{Co}(\text{OH})_2$ after soaking for 12 and 24 h. The reconfiguration rate of $\text{CoC}_2\text{O}_4@\text{MXene}$ is slow and cannot be completely converted to $\text{Co}(\text{OH})_2$ without applying potentials, indicating that the potential is one of the important reasons for triggering the fast complete reconfiguration. In addition, no reconfiguration phenomenon is found in $\text{CoC}_2\text{O}_4@\text{MXene}$ after neutral HER on $\text{CoC}_2\text{O}_4@\text{MXene}$ by XRD analysis. The above results indicate that applying a potential

under alkaline electrolyte etching can promote the rapid and complete reconfiguration of the catalyst. The corresponding analysis was added in the revised manuscript. Please see Line 17-19, Page 8, highlighted in yellow. As a brief description, R-CoC₂O₄@MXene is prepared using the alkaline HER process as a catalyst synthesis method, and then the HER performance is examined in neutral seawater.

Supplementary Fig. 22. a) XRD patterns of CoC₂O₄@MXene after soaking in 1M KOH for 6, 12, and 24 h, respectively. b) XRD patterns of CoC₂O₄@MXene after neutral HER.

5) In Supplementary Figure 11, please define crystal growth direction and exposed planes of Co(OH)₂@MXene assisted by electron diffraction patterns and d-spacing. Then compare with those of pristine MXene and as-prepared Co(OH)₂@MXene in the Supplementary Figure 17)

Our response: We thank the reviewer for the good comment. The TEM images of R-CoC₂O₄@MXene show that the hexagonal Co(OH)₂ nanosheets are tightly connected with MXene (Supplementary Fig. 15). As shown in Fig. 2c, the lattice fringe with an interplanar spacing of 0.276 nm is indexed into the (100) plane of Co(OH)₂, indicating that the hexagonal Co(OH)₂ nanosheets mainly expose the {001} plane. Because the growth direction of Co(OH)₂ nanosheets belongs to the two-dimensional extension type, the growth rate is faster in the [100] direction. Moreover, the growth of Co(OH)₂ nanosheets along the [001] direction is unstable due to the lack of chemical bonds. The above results indicate that the Co(OH)₂ nanosheets mainly grow along the [100]

direction during the reconfiguration process. Meanwhile, compared with the pristine MXene (Supplementary Fig. 6), the MXene in R-CoC₂O₄@MXene still maintains the ultra-thin sheet-like structure, which further confirms the superior stability of MXene during the HER process. In addition, the as-prepared Co(OH)₂@MXene exhibits a regular hexagonal sheet structure with a well-defined outline and transparent features, in which the ultrathin MXene adheres to the Co(OH)₂ nanosheets (Supplementary Fig. 19a). The HRTEM image shows the lattice distance of 0.276 nm (Supplementary Fig. 19b), corresponding to the (100) plane in Co(OH)₂. The selected area electron diffraction (SAED) pattern (Supplementary Fig. 19c) presents the diffraction spots for (100) and (110) planes from Co(OH)₂. This indicates that both R-CoC₂O₄@MXene and the as-prepared Co(OH)₂@MXene grow along the [100] direction and expose the {001} plane. Interestingly, compared with as-prepared Co(OH)₂@MXene, the lattice fringes of R-CoC₂O₄@MXene are blurred and weakened, which can be attributed to the formation of oxygen defects. Therefore, the R-CoC₂O₄@MXene obtained by electrochemical reconfiguration can expose a large number of unsaturated Co active sites to enhance the HER performance. The corresponding analysis was added in the revised manuscript. Please see Line 22-23, Page 7 and Line 1-5, Page 8, highlighted in yellow.

Supplementary Fig. 15. a) Low-magnified and b) high-magnified TEM images of the R-CoC₂O₄@MXene after cycling, respectively.

Fig. 2. c) HRTEM image of R-CoC₂O₄@MXene.

Supplementary Fig. 6. a) TEM and b, c) HRTEM images of MXene.

Supplementary Fig. 19. a) TEM and b) HRTEM images of as-prepared Co(OH)₂@MXene. c) SAED pattern of as-prepared Co(OH)₂@MXene.

6) It would be good that the authors provide more EIS data with an insert for smaller ohm regions and a detail discussion on charge/mass transfers matching with materials characterization.

Our response: We thank the reviewer for the valuable suggestion. According to your suggestion, we have inserted smaller ohmic regions and fitted the Nyquist plots of R-CoC₂O₄@MXene, R-CoC₂O₄, MXene, and Pt/C. The corresponding fitting parameters

are listed in Supplementary Table 3 and 6. As shown in the Nyquist plots (Supplementary Fig. 30), the EIS spectrum of R-CoC₂O₄@MXene in 1 M KOH electrolyte displays a characteristic semicircle with a very small diameter, indicating low charge transfer resistance (R_{ct}). The charge transfer resistance of R-CoC₂O₄@MXene (2.370) is much lower than that of R-CoC₂O₄ (9.732) and MXene (36.832), indicating that R-CoC₂O₄@MXene has efficient electron transfer kinetics during HER process. Meanwhile, R-CoC₂O₄@MXene also exhibits superior charge transfer rates in 1 M KOH seawater and neutral seawater (Supplementary Fig. 44). The superior charge transfer is attributed to the combination of MXene with high electrical conductivity with low-valence Co(OH)₂, which forms an efficient electron transport channel. The corresponding analysis has been added to the revised manuscript, please see Line 11-14, Page 11, highlighted in yellow.

Supplementary Fig. 30. The EIS curves for HER of MXene, R-CoC₂O₄, R-CoC₂O₄@MXene, and Pt/C in 1 M KOH, respectively.

Supplementary Fig. 44. Nyquist plots of R-CoC₂O₄@MXene electrodes in a) 1 M KOH seawater and b) neutral seawater, respectively.

Supplementary Table 3. The fitting impedance parameters of MXene, R-CoC₂O₄, R-CoC₂O₄@MXene, and Pt/C in 1 M KOH, respectively.

Samples	R _s	R1	CPE1		R2	CPE2	
			CPE1-T	CPE1-P		CPE2-T	CPE2-P
R-CoC ₂ O ₄ @MXene	1.029	0.469	0.047	0.547	2.370	0.109	0.915
R-CoC ₂ O ₄	1.130	0.846	0.011	0.621	9.732	0.014	0.967
MXene	1.832	1.620	0.021	0.497	36.832	0.022	0.870
Pt/C	1.172	0.358	0.043	0.592	2.279	0.156	0.877

Supplementary Table 6. The fitting impedance parameters of MXene, R-CoC₂O₄, R-CoC₂O₄@MXene, and Pt/C in 1 M KOH seawater and neutral seawater, respectively.

Samples	R _s	R1	CPE1	
			CPE1-T	CPE1-P

R-CoC ₂ O ₄ @MXene (1 M KOH seawater)	1.034	17.764	0.146	0.784
Pt/C	1.125	35.236	0.156	0.569
R-CoC ₂ O ₄ @MXene (1 M KOH seawater)	0.958	68.412	0.126	0.657
Pt/C	1.021	102.351	0.115	0.548

7) Minor revisions such as Line 110 scan electron microscopy (SEM), Supplementary Figure 13 without y-axis (Potential) scale, Supplementary Figure 14 with missing indications in R-CoC₂O₄@MXene (e.g., lower than 400 eV), and representing CV after 3000 cycles (text line 266) to Supplementary Figure 23 (polarization curves)

Our response: We thank the reviewer for the good comments. “scan electron microscopy (SEM)” has been revised to “scanning electron microscopy (SEM)”. The y-axis (potential) scale of the potential-dependent in situ Raman spectra of R-CoC₂O₄@MXene has been added. Indications below 400 eV have been added to the full XPS survey spectra, where Cl 2p, K 2p, and K 2s are introduced during MAX etching and HER detection, respectively. In addition, we also supplement the CV with 3000 cycles. Please see Supplementary Fig. 20, 23, and 34 in the revised manuscript, highlighted in yellow.

Supplementary Fig. 20. Potential-dependent in situ Raman spectra of R-

CoC₂O₄@MXene.

Supplementary Fig. 23. Comparison of full XPS survey spectra of CoC₂O₄, CoC₂O₄@MXene, and R-CoC₂O₄@MXene.

Supplementary Fig. 34. a) 3000 cycles CV of R-CoC₂O₄@MXene. b) Polarization curves of R-CoC₂O₄@MXene at the initial stage and after 3000 cycles.

At last, we wish to thank the Reviewers and the Editors again for the constructive comments and suggestions to improve the quality of our manuscript. Thank you very much!

REVIEWERS' COMMENTS

Reviewer #1 (Remarks to the Author):

The authors have addressed all my comments and I believe this study merits publication in Nature Comm.

Reviewer #2 (Remarks to the Author):

The manuscript has been improved. However, there are still a few issues to be addressed.

1. Regarding to the reply to #1 of reviewer 2, the authors must understand that hydrogen is NOT an energy source, but a carrier. Hydrogen has to be produced from a primary energy source!
2. The reconstruction mechanism of $\text{CoC}_2\text{O}_4@\text{MXene}$ to $\text{R-CoC}_2\text{O}_4@\text{MXene}$ is not convincing. Over potentials starts decreasing over the LSV scans from 74 mV to 28 mV. However, one or two cycles of LSV are not enough to conclude the saturation point. The authors may need to perform a CP study for one hour at the saturated potential to confirm further reconstruction below 28 mV. Is the Co(OH)_2 stable after 100 hours of electrochemical reaction? And is there any TiO_2 formation after the electrochemical reaction?
3. It is great that the comparison of the overpotential and current density was provided. It may add more values to also include the stability of catalysts to provide insight for practical implementation.

Reviewer #3 (Remarks to the Author):

The authors diligently performed the revision process providing lots of data and explanations in the response letter. However, revised contents have not been well reflected in the revised manuscript, therefore, the reviewer suggests more organizing the correlation between provided data and explanation in the manuscript before being published in Nature Communications. Here are a few more suggestions for the authors.

- 1) First of all, the authors should include additional information relevant to the experiment performed in the manuscript. The revised manuscript has not provided detail in Methods such as preparation of Co(OH)_2 , $\text{Co(OH)}_2@\text{MXene}$ and characterization of contact angle measurements
- 2) The electron diffraction patterns of a monocrystal show discrete dots indicating each crystal plane that is perpendicular to the zone axis (i.e. incident beam direction) [Williams, D. B. & Carter, C. B. Transmission electron microscopy: a textbook for materials science. (Springer, 2008).] Therefore, electron diffraction (e.g. Supplementary Figure 19c) should be carefully addressed instead of covering all

the dots as a ring of (100). Moreover, the author cannot say “The lattice fringes of 0.276 nm correspond to the (100) plane of Co(OH)_2 species (Supplementary Fig. 19b). This result indicates that the crystal growth directions of the Co(OH)_2 species in $\text{R-CoC}_2\text{O}_4@\text{MXene}$ and $\text{Co(OH)}_2@\text{MXene}$ are consistent.” Corresponding lattice data ensure the crystal phase of the materials, but not the growth direction nor surface exposed planes. Please carefully compare $\text{R-CoC}_2\text{O}_4@\text{MXene}$ and chemically prepared $\text{Co(OH)}_2@\text{MXene}$ by HR-TEM including the outmost surface as the active area and interface between MXene either $\text{R-CoC}_2\text{O}_4$ or Co(OH)_2 . Then please correlate plenty of unsaturated Co in $\text{R-CoC}_2\text{O}_4$ (i.e. not in chemically prepared Co(OH)_2) as the spectroscopic explanations.

3) As the author explained in the response letter, “Herein, the complete reconfiguration of the crystal phase has been achieved by the rational design of the catalyst. We refer to the crystal phase transition from $\text{CoC}_2\text{O}_4@\text{MXene}$ to $\text{Co(OH)}_2@\text{MXene}$ as complete reconfiguration..... The reconfiguration rate of $\text{CoC}_2\text{O}_4@\text{MXene}$ is slow and cannot be completely converted to Co(OH)_2 without applying potentials, indicating that the potential is one of the important reasons for triggering the fast complete reconfiguration.” As Supplementary figure 22 indicates alkali medium (i.e. storing sample in 1M KOH) is more important than potentials (i.e. neutral HER). Does electric potential matter fast reconfiguration rather than any other effects (e.g., defect)? To confirm the significance of reconfiguration under the chemical and electrochemical treatment, please compare the electronic and microscopic structures between “suggested $\text{R-CoC}_2\text{O}_4@\text{MXene}$ ” and “ $\text{CoC}_2\text{O}_4@\text{MXene}$ after soaking in 1M KOH for more than 24 h”.

4) Still, there are some errors and missing things; especially in supplementary figure 7 “After the combination of CoC_2O_4 and MXene, the zeta potential of $\text{CoC}_2\text{O}_4@\text{MXene}$ (+10.37 mV) is positively shifted compared with MXene.” 10.37mV in green color represents “ CoC_2O_4 ”, not $\text{CoC}_2\text{O}_4@\text{MXene}$. The blue color represents “ $\text{R-CoC}_2\text{O}_4@\text{MXene}$ ” in the data but there is no $\text{R-CoC}_2\text{O}_4@\text{MXene}$ in the caption and explanations. Please be careful that all the samples and data should be identical to each other because the authors emphasized the significance of complete reconfiguration assisted by alkali etching and electrochemical methods. $\text{CoC}_2\text{O}_4@\text{MXene}$ and $\text{R-CoC}_2\text{O}_4@\text{MXene}$ (i.e. defective $\text{Co(OH)}_2@\text{MXene}$, also different from chemically as-prepared $\text{Co(OH)}_2@\text{MXene}$) are totally different in this work. “Natural” in supplementary figure 22b looks like “Neutral” seawater.

Point-by-Point Response to Review Comments

We sincerely appreciate the pertinent suggestions of editors and reviewers. The quality of the manuscript has been significantly improved following reviewers' comments. All modifications have been highlighted with yellow shading in the revised manuscript.

Reviewers' comments:

Reviewer #1(Remarks to the Author): The authors have addressed all my comments and I believe this study merits publication in Nature Comm.

Our response: We thank the reviewer for acknowledging the acceptance of our work.

Reviewer #2 (Remarks to the Author): The manuscript has been improved. However, there are still a few issues to be addressed.

1. Regarding to the reply to #1 of reviewer 2, the authors must understand that hydrogen is NOT an energy source, but a carrier. Hydrogen has to be produced from a primary energy source!

Our response: We thank the reviewer for the good comment. We agree that hydrogen is not an energy source, but a carrier, and must be produced from a primary energy source. The description of " Hydrogen fuel is the most promising alternative to fossil energy" has been corrected in the manuscript. Please see the introduction section in the revised manuscript. We thank you again for your positive comments and valuable suggestions to improve the quality of our manuscript.

2. The reconstruction mechanism of $\text{CoC}_2\text{O}_4@\text{MXene}$ to $\text{R-CoC}_2\text{O}_4@\text{MXene}$ is not convincing. Over potentials starts decreasing over the LSV scans from 74 mV to 28 mV. However, one or two cycles of LSV are not enough to conclude the saturation point. The authors may need to perform a CP study for one hour at the saturated potential to confirm further reconstruction below 28 mV. Is the Co(OH)_2 stable after

100 hours of electrochemical reaction? And is there any TiO_2 formation after the electrochemical reaction?

Our response: We thank the reviewer for the significant comment. According to your suggestion, we have performed chronopotentiometry measurement of the transformation of $\text{CoC}_2\text{O}_4@\text{MXene}$ to $\text{R-CoC}_2\text{O}_4@\text{MXene}$. As shown in Supplementary Fig. 11, the catalyst can maintain 28 mV during the 1 h chronopotentiometry test after a short activation process. Therefore, the catalyst remained stable without further reconstruction at 28 mV. The structure of $\text{R-CoC}_2\text{O}_4@\text{MXene}$ after 100 h of electrochemical reaction was analyzed by XRD diffraction. The diffraction peaks of $\text{R-CoC}_2\text{O}_4@\text{MXene}$ can be well assigned to the Co(OH)_2 phase after 100 h of electrochemical reaction (Supplementary Fig. 38), demonstrating the robust structure of $\text{R-CoC}_2\text{O}_4@\text{MXene}$. In addition, the formation of TiO_2 phase was not found. The corresponding analysis was added in the revised manuscript. Please see Line 27-28, Page 5 and Line 21-23, Page 10, highlighted in yellow.

Supplementary Fig. 11. Chronopotentiometry curves of $\text{CoC}_2\text{O}_4@\text{MXene}$ at constant current densities of 10 mA cm^{-2} .

Supplementary Fig. 38. XRD of R-CoC₂O₄@MXene after stability.

3. It is great that the comparison of the overpotential and current density was provided. It may add more values to also include the stability of catalysts to provide insight for practical implementation.

Our response: We thank the reviewer for the significant comment. Based on your suggestion, we have supplemented the stability comparison of catalysts to better evaluate the practical implementation value. As shown in Supplementary Table S1 and S2, the stability of R-CoC₂O₄@MXene is superior to that of most catalysts. We thank you again for your positive comments and valuable suggestions to improve the quality of our manuscript.

Supplementary Table 1. Summary of reconfiguration product, reconfiguration degree, reconfiguration time, and HER performance of different pre-catalysts in 1 M KOH.

Pre-catalyst	Reconfiguration product	Reconfiguration degree	Reconfiguration time	Overpotential (mV) 10 mA cm ⁻²	Stability (h)	References
CoC ₂ O ₄ @MXe	Co(OH) ₂ @MXe	Complete reconfiguration	5 cycles LSV (8.3 min)	28	100@10 mA cm ⁻²	This Work

Ni ₄ Mo	Ni ₄ Mo@Mo ₂ O ₇ ²⁻	Surface reconfiguration	4 h	30	50@58 mA cm ⁻²	Angew. Chem. Int. Ed. 2021, 60, 7051-7055.
NiSe ₂	NiSe/NiSe ₂	Deep reconfiguration	20 cycles LSV	50	25@160 mV vs RHE	ACS Energy Lett. 2020, 5, 2483-2491.
TiO ₂ Co ₂ P ₄ O ₁₂	CoO _x (OH) _y /TiO ₂ Co ₂ P ₄ O ₁₂	Surface reconfiguration	35 cycles LSV	50	40@90 mV vs RHE	J. Mater. Chem. A. 2019, 7, 12457-12467.
CoF ₂	Co(OH) ₂	Complete reconfiguration	60 cycles LSV	54	110@10 mA cm ⁻²	Adv. Sci. 2022, 9, 2103567.
Co _{3-x} Ni _x O ₄	Co _y Ni _{1-y} O/Co _{3-x} Ni _x O ₄	Surface reconfiguration	20 h	57	300@20 mA cm ⁻²	ACS Catal. 2021, 11, 8174-8182.
Mo ₂ C- Mo(VI)O _x /CC	Mo ₂ C- Mo(IV/VI)O _x /CC	Surface reconfiguration	100 cycles LSV	60	25@200 mV vs RHE	Angew. Chem. Int. Ed. 2020, 59, 3544-3548.
TiO ₂ @CoCH	Co(OH) ₂ /TiO ₂ @ CoCH	Surface reconfiguration	25 cycles LSV	66	40@50 mV vs RHE	Nano Energy. 2021, 82, 105732.
NF-NiS ₂	Ni/NF-NiS ₂	Surface reconfiguration	100 cycles LSV	67	40@20 mA cm ⁻²	Nano Energy. 2017, 41, 148-153.
Ni-BDT	NiNSs/ Ni-BDT	Surface reconfiguration	1000 cycles LSV	80	30@20 mA cm ⁻²	Chem. 2017, 3, 122-133.
CoFeO@BP	CoP/CoFeO@BP	Surface reconfiguration	20 cycles LSV	88	22.5@90 mV vs RHE	Angew. Chem. Int. Ed. 2020, 59, 21106-21113.
CoSe _{1.26} P _{1.42}	Co/CoSe _{1.26} P _{1.42}	Deep reconfiguration	N.A.	92	15@92 mV vs RHE	ACS Energy Lett. 2019, 4, 987-994.
CoP	Co(OH) _x /CoP	Surface reconfiguration	10 h	100	25@92 mV vs RHE	Chem. Sci. 2019, 10, 2019-2024.

$\text{Co}_{0.5}\text{W}_{0.5}\text{S}_x$	$\text{CoO/Co(OH)}_2/\text{Co}_{0.5}\text{W}_{0.5}\text{S}_x$	Surface reconfiguration	10 h	110	10	J. Mater. Chem. A. 2021, 9, 11359-11369.
CoNP@C	$\text{Co(OH)}_2/\text{CoNP@C}$	Surface reconfiguration	3 h	191	20@20 mA cm^{-2}	Nano Energy. 2018, 49, 14-22.

Supplementary Table 5. Comparison of HER activity measured for $\text{R-CoC}_2\text{O}_4\text{@MXene}$ with other representative reported HER catalysts using 1.0 M KOH as electrolyte.

Sample	Overpotential (mV) 10 mA cm^{-2}	Overpotential (mV) 500 mA cm^{-2}	Overpotential (mV) 1000 mA cm^{-2}	Stability (h)	References
R-CoC₂O₄@MXene	28	157	216	100@10, 500, 1000 mA cm^{-2}	This work
2H Nb _{1.35} S ₂	46	247	370	120@1000 mA cm^{-2}	Nat Mater. 2019, 18, 1309-1314.
W ₁ Mo ₁ -NG DAC	67	N.A.	N.A.	27@10 mA cm^{-2}	Sci. Adv. 2020, 6, eaba6586.
NFP/C-3	95	N.A.	N.A.	12@10 mA cm^{-2}	Sci. Adv. 2019, 5, eaav6009.
Fe(OH) _x @Cu-MOF NBs	112	N.A.	N.A.	30@112 mV vs RHE	Sci. Adv. 2021, 7, eabg2580.
NiMoO _x /NiMoS	38	186	240	25@500 mA cm^{-2}	Nat. Commun. 2020, 11, 5462.
MoS ₂ /Mo ₂ C	45	191	220	24@200 mA cm^{-2}	Nat. Commun. 2019, 10, 269.
FeP/Ni ₂ P	85	230	265	24@100 mA cm^{-2}	Nat. Commun. 2018, 9, 2551.
CF/VMFP	45	230	N.A.	30@250 mA cm^{-2}	Nat. Commun. 2021, 12, 1380.
MoC-Mo ₂ C-790	98.2	292	N.A.	50@500 mA cm^{-2}	Nat. Commun. 2021, 12, 6776.

HC-MoS ₂ /Mo ₂ C	280	320	412	24@400 mV vs RHE	Nat Commun. 2020, 11, 3724.
PW-Co ₃ N NWA/NF	41	N.A.	N.A.	25@92 mV vs RHE	Nat. Commun. 2020, 11, 1853.
Ni-Fe NP	46	N.A.	N.A.	24@10 mA cm ⁻²	Nat. Commun. 2019, 10, 5599.
NiCo-SAD-NC	61	N.A.	N.A.	15@100 mA cm ⁻²	Nat Commun. 2021, 12, 6766.
3CoMo-Vs	75	N.A.	N.A.	10@10 mA cm ⁻²	Nat Commun. 2020, 11, 2253.
1T _{0.81} -MoS ₂ @NiS ₂	95	N.A.	N.A.	16@10 mA cm ⁻²	Nat Commun. 2021, 12, 5260.
Co-doped CeO ₂	75	240	N.A.	14@150 and 200 mV vs RHE	J. Am. Chem. Soc. 2020, 142, 6461-6466.
Ni ₂ P/NF	85	245	300	10@2500 mA cm ⁻²	J. Am. Chem. Soc. 2019, 141, 7537-7543.
Co-NC-AF	85	272	343	32@1000 mA cm ⁻²	Adv.Mater. 2021, 33, 2103533.
P-Fe ₃ O ₄ /IF	39	212	242	20@100, 300, and 500 mA cm ⁻²	Adv. Mater. 2019, 31, 1905107.
Cr _{0.4} Mo _{0.6} B ₂	180	390	N.A.	25@200 mV vs RHE	Adv. Mater. 2020, 32, 2000855.
CuCo-CAT/CC	52	N.A.	N.A.	10@10 mA cm ⁻²	Adv. Mater. 2021, 33, 2106781.
CoNi-inf	72	N.A.	N.A.	14@10 mA cm ⁻²	Adv. Mater. 2020, 32, 2002857.

Ni/Ni(OH) ₂	77	N.A.	N.A.	10@10 mA cm ⁻²	Adv. Mater. 2020, 32, 1906915.
Ni@N-HCGHF	95	N.A.	N.A.	10@10 mA cm ⁻²	Adv. Mater. 2020, 32, 2003313.
MoO ₂ -FeP@C	103	N.A.	N.A.	24@10 mA cm ⁻²	Adv. Mater. 2020, 32, 2000455.
Ni ₄ Mo	34	N.A.	N.A.	50@58 mA cm ⁻²	Angew. Chem. Int. Ed. 2021, 60, 7051-7055.
Ni ₃ N-Co ₃ N PNAs/NF	43	N.A.	N.A.	40@25 mA cm ⁻²	Angew. Chem. Int. Ed. 2021, 60, 5984-5993.
S-NiFe ₂ O ₄	61	N.A.	N.A.	60@200 mA cm ⁻²	Angew. Chem. Int. Ed. 2021, 60, 14117-14123.
Co-MoS ₂ @CoS ₂	76	N.A.	N.A.		Angew. Chem. Int. Ed. 2022, 61, e2021148.
CoFeO@BP	88	N.A.	N.A.	22.5@10 mA cm ⁻²	Angew. Chem. Int. Ed. 2020, 59, 21106-21113.
P-MoP/Mo ₂ N	89	N.A.	N.A.	48@20 mA cm ⁻²	Angew. Chem. Int. Ed. 2021, 60, 6673-6681.
CoP _x @CNS	91	N.A.	N.A.	60@16 mA cm ⁻²	Angew. Chem. Int. Ed. 2020, 59, 21360-21366.
N-c-CoSe ₂	98	N.A.	N.A.	24@130 mV vs RHE	Angew. Chem. Int. Ed. 2021, 60, 21575-21582.

S-Co _{0.85} Se-1	108	N.A.	N.A.	48@10 mA cm ⁻²	Angew. Chem. Int. Ed. 2021, 60, 12360-12365.
2D meso-Mo ₂ C/Mo ₂ N	110	N.A.	N.A.	20@10 mA cm ⁻²	Angew. Chem. Int. Ed. 2022, 134, e202112298.
NiP ₂ -650	134	N.A.	N.A.	14@50 mA cm ⁻²	Angew. Chem. Int. Ed. 2021, 60, 259-267.
Co-MoS ₂	137	N.A.	N.A.	N.A.	Angew. Chem. Int. Ed. 2021, 60, 7251-7258.
Co _{0.9} Ni _{0.1} Se	185.7	N.A.	N.A.	N.A.	Angew. Chem. Int. Ed. 2020, 59, 22743-22748.
Co-Mo ₅ N ₆	19	206	280	14@50 mA cm ⁻²	Adv. Energy Mater. 2020, 10, 2002176.
C-Ni _{1-x} O/3DPNi	32	220	245	10@10, 50, 1000 mA cm ⁻²	Adv. Energy Mater. 2020, 10, 2002955.
Ni/FeOOH	38	287	N.A.	18@100 mA cm ⁻²	Adv. Energy Mater. 2020, 10, 1904020.
NFN-MOF/NF	87	293	N.A.	30@250, 500 mA cm ⁻²	Adv. Energy Mater. 2018, 8, 1801065.
E-Co SAs	59	280	N.A.	500@10 mA cm ⁻²	Adv. Funct. Mater. 2021, 31, 2100547.
Ni-Mo-B HF	23	329	N.A.	300@450 mA cm ⁻²	Adv. Funct. Mater. 2021, 32, 2107308.

						Adv. Funct.
PS-Cu	121	750	980	30@100 mA cm ⁻²		Mater. 2022, 32, 2112367.
NiMnOP/NF	81	195	N.A.	35@500 mA cm ⁻²		Nano Energy. 2020, 69, 104432.
NiFe-LDH/MXene/NF	132	205	N.A.	280@10 mA cm ⁻²		Nano Energy. 2019, 63, 103880.
MFN-MOFs(2:1)/NF	79	234	N.A.	100@500 mA cm ⁻²		Nano Energy. 2019, 57, 1-13.
Ni ₂ P-CuP ₂	51	360	460	75@100, 200, 500 mA cm ⁻²		ACS Nano. 2021, 15, 5586-5599.
Ni ₃ Sn ₂ S ₂ @Ni ₃ S ₂	50	206	N.A.	20@200 mA cm ⁻²		Appl. Catal. B: Environ. 2020, 267, 118675.
NiFeMo IOS@NF	33	249	N.A.	50@500 mA cm ⁻²		Appl. Catal. B: Environ. 2020, 267, 118376.
A-NiCo LDH/NF	43	286	381	72@291, 381 mV vs RHE		Appl. Catal. B: Environ. 2020, 261, 118240.
IrFe/NC	37	510	850	12@100 mA cm ⁻²		Appl. Catal. B: Environ. 2019, 258, 117965.
NF@NiFeLDH	166	356	N.A.	120@300 mA cm ⁻²		Small. 2021, 18, 2104354.

Reviewer #2 (Remarks to the Author): The authors diligently performed the revision process providing lots of data and explanations in the response letter. However, revised contents have not been well reflected in the revised manuscript, therefore, the reviewer suggests more organizing the correlation between provided data and explanation in the manuscript before being published in Nature Communications. Here are a few more suggestions for the authors.

1. First of all, the authors should include additional information relevant to the experiment performed in the manuscript. The revised manuscript has not provided detail in Methods such as preparation of Co(OH)_2 , $\text{Co(OH)}_2\text{@MXene}$, and characterization of contact angle measurements.

Our response: We thank the reviewer for the valuable comment. Based on your suggestion, we have supplemented method details for the preparation of Co(OH)_2 and $\text{Co(OH)}_2\text{@MXene}$ as well as the contact angle measurements. Please see the Methods and Characterizations section in the revised manuscript.

2. The electron diffraction patterns of a monocrystal show discrete dots indicating each crystal plane that is perpendicular to the zone axis (i.e. incident beam direction) [Williams, D. B. & Carter, C. B. Transmission electron microscopy: a textbook for materials science. (Springer, 2008).]. Therefore, electron diffraction (e.g. Supplementary Figure 19c) should be carefully addressed instead of covering all the dots as a ring of (100). Moreover, the author cannot say “The lattice fringes of 0.276 nm correspond to the (100) plane of Co(OH)_2 species (Supplementary Fig. 19b). This result indicates that the crystal growth directions of the Co(OH)_2 species in R- $\text{CoC}_2\text{O}_4\text{@MXene}$ and $\text{Co(OH)}_2\text{@MXene}$ are consistent.” Corresponding lattice data ensure the crystal phase of the materials, but not the growth direction nor surface exposed planes. Please carefully compare R- $\text{CoC}_2\text{O}_4\text{@MXene}$ and chemically prepared $\text{Co(OH)}_2\text{@MXene}$ by HR-TEM including the outmost surface as the active area and interface between MXene either R- CoC_2O_4 or Co(OH)_2 . Then please correlate plenty of unsaturated Co in R- CoC_2O_4 (i.e. not in chemically prepared Co(OH)_2) as the spectroscopic explanations.

Our response: We thank the reviewer for the valuable comment. We have carefully dealt with the electron diffraction in Supplementary Fig. 20e. We agree with you and have revised the description of the catalyst growth direction and exposed planes. In addition, we have carefully calibrated the electron diffraction pattern. To further explore the microstructure of R- $\text{CoC}_2\text{O}_4\text{@MXene}$ and as-prepared $\text{Co(OH)}_2\text{@MXene}$, HRTEM of the outmost surface as the active area and interface between MXene and R-

CoC₂O₄ or Co(OH)₂ are provided. As shown in Supplementary Fig. 16c and 20b, the lattice fringe with an interplanar spacing of 0.276 nm is indexed to the (100) plane of Co(OH)₂, while a d-spacing of around 0.950 nm corresponds to the typical interlayer of MXene. This indicates the formation of a tightly connected heterostructure between MXene and R-CoC₂O₄ or Co(OH)₂. Supplementary Fig. 16d and 20c show HRTEM images of Co(OH)₂ hexagonal nanosheets, in which the three types of lattice fringes correspond to interplanar spacings of 0.276 nm. The angles between any of them are 60°. These planes are consistent with the SAED pattern projection of Co(OH)₂ along the [001] direction (Supplementary Fig. 16f and 20e). According to the crystal structure of Co(OH)₂, the ($\bar{1}10$), (010), and (100) planes are equivalent. Therefore, the (001) and (100) planes are exposed on the hexagonal Co(OH)₂ nanosheets (Supplementary Fig. 20f). Moreover, Supplementary Fig. 16e and 20d show that the side-view HRTEM images of Co(OH)₂ nanosheet can clearly observe well-resolved lattice fringes. The interlayer spacing of 0.466 nm corresponds to the (001) plane of Co(OH)₂. The above results indicate that the Co(OH)₂ in R-CoC₂O₄@MXene and the as-prepared Co(OH)₂@MXene have consistent crystal structures. As the growth direction of Co(OH)₂ nanosheets belongs to the two-dimensional extension type, the growth rate is faster in the <100> direction. Therefore, the Co(OH)₂ in R-CoC₂O₄@MXene and as-prepared Co(OH)₂@MXene grows along the <100> direction and is mainly exposed to {001} planes. Moreover, the HRTEM of Co(OH)₂ in R-CoC₂O₄@MXene exhibits more lattice defects compared to the as-prepared Co(OH)₂@MXene, which is also consistent with the XAS analysis results. The corresponding analysis has been added in the revised manuscript. Please see Line 1-5, Page 7, Supplementary Fig. 16, and Supplementary Fig. 20 highlighted in yellow.

Supplementary Fig. 16. a, b) Low-magnified images, c-e) high-magnified TEM images, and f) SAED pattern of the R-CoC₂O₄@MXene after cycling, respectively.

Supplementary Fig. 20. a) Low-magnified image, b-d) high-magnified TEM images, and e) SAED pattern of the as-prepared Co(OH)₂@MXene, respectively. f) Simulated crystal morphology of Co(OH)₂.

3. As the author explained in the response letter, “Herein, the complete reconfiguration of the crystal phase has been achieved by the rational design of the catalyst. We refer to the crystal phase transition from CoC₂O₄@MXene to Co(OH)₂@MXene as complete reconfiguration..... The reconfiguration rate of CoC₂O₄@MXene is slow and cannot

be completely converted to Co(OH)_2 without applying potentials, indicating that the potential is one of the important reasons for triggering the fast complete reconfiguration.” As Supplementary figure 22 indicates alkali medium (i.e. storing sample in 1M KOH) is more important than potentials (i.e. neutral HER). Does electric potential matter fast reconfiguration rather than any other effects (e.g., defect)? To confirm the significance of reconfiguration under the chemical and electrochemical treatment, please compare the electronic and microscopic structures between “suggested R- CoC_2O_4 @MXene” and ‘ CoC_2O_4 @MXene after soaking in 1M KOH for more than 24 h”.

Our response: We thank the reviewer for the valuable comment. The XRD patterns of CoC_2O_4 @MXene after soaking in 1M KOH for 24 h clearly show that the main peak is still consistent with CoC_2O_4 , and only a few characteristic peaks correspond to Co(OH)_2 species (Supplementary Fig. 23a). As comparisons, CoC_2O_4 @MXene is completely converted to Co(OH)_2 @MXene and the reconfiguration time is greatly shortened under electrochemical HER conditions (Supplementary Fig. 14). This result verifies that the electric potential has a significant effect on the degree and time of reconfiguration. In addition, no reconfiguration phenomenon is found in CoC_2O_4 @MXene after neutral HER on CoC_2O_4 @MXene by XRD analysis (Supplementary Fig. 23b). Therefore, both the potential and the alkaline electrolyte appear to play a significant role in the rapid and complete reconfiguration of the catalyst.

To confirm the importance of reconfiguration under chemical and electrochemical treatments, the electronic and microscopic structures of R- CoC_2O_4 @MXene and CoC_2O_4 @MXene after soaking in 1M KOH for 24 h were further explored. R- CoC_2O_4 @MXene exhibits a large number of hexagonal nanosheets interconnected to form a rod-like loose structure after electrochemical reconfiguration (Supplementary Fig. 15). The surface of CoC_2O_4 @MXene after immersion in 1 M KOH solution for 24 h only exhibits the formation of sparse Co(OH)_2 nanosheets (Supplementary Fig. 24). This indicates that CoC_2O_4 @MXene tends to undergo complete reconfiguration under the applied potential. The Co 2p spectrum of CoC_2O_4 @MXene soaked in 1M KOH for 24 h in Supplementary Fig. 27 shows that the characteristic peaks of Co 2p_{3/2} and Co

$2p_{1/2}$ can be observed at binding energies of 777.18 and 793.26 eV with two satellite peaks at 783.23 and 799.45 eV. Obviously, the binding energies of Co $2p_{3/2}$ and Co $2p_{1/2}$ for R-CoC₂O₄@MXene are lower than of CoC₂O₄@MXene soaked in 1M KOH for 24 h, indicating that the Co atoms in R-CoC₂O₄@MXene are reduced to lower valence states. The corresponding analysis has been added in the revised manuscript. Please see Line 20-22, Page 7 and Line 4-6, Page 8, highlighted in yellow.

Supplementary Fig. 23. a) XRD patterns of CoC₂O₄@MXene after soaking in 1M KOH for 6, 12, and 24 h, respectively. b) XRD patterns of CoC₂O₄@MXene after neutral HER.

Supplementary Fig. 14. XRD patterns of R-CoC₂O₄@MXene and R-CoC₂O₄, respectively.

Supplementary Fig. 15. a) Low-magnified and b) high-magnified SEM images of R-CoC₂O₄@MXene after cycling, respectively.

Supplementary Fig. 24. a) Low-magnified and b) high-magnified SEM images of CoC₂O₄@MXene soaked in 1 M KOH for 24 h, respectively.

Supplementary Fig. 27. a) Full XPS survey spectrum of CoC₂O₄@MXene soaked in 1 M KOH for 24 h. b) XPS spectra of the Co 2p region of R-CoC₂O₄@MXene and CoC₂O₄@MXene soaked in 1 M KOH for 24 h, respectively.

4. Still, there are some errors and missing things; especially in supplementary figure 7 “After the combination of CoC_2O_4 and MXene, the zeta potential of $\text{CoC}_2\text{O}_4@\text{MXene}$ (+10.37 mV) is positively shifted compared with MXene.” 10.37 mV in green color represents “ CoC_2O_4 ”, not $\text{CoC}_2\text{O}_4@\text{MXene}$. The blue color represents “ $\text{R-CoC}_2\text{O}_4@\text{MXene}$ ” in the data but there is no $\text{R-CoC}_2\text{O}_4@\text{MXene}$ in the caption and explanations. Please be careful that all the samples and data should be identical to each other because the authors emphasized the significance of complete reconfiguration assisted by alkali etching and electrochemical methods. $\text{CoC}_2\text{O}_4@\text{MXene}$ and $\text{R-CoC}_2\text{O}_4@\text{MXene}$ (i.e. defective $\text{Co}(\text{OH})_2@\text{MXene}$, also different from chemically as-prepared $\text{Co}(\text{OH})_2@\text{MXene}$) are totally different in this work. “Natural” in supplementary figure 22b looks like “Neutral” seawater.

Our response: We thank the reviewer for the good comment. We have revised the corresponding catalyst labels in Supplementary Fig. 7. Blue, green, and orange correspond to $\text{CoC}_2\text{O}_4@\text{MXene}$, CoC_2O_4 , and MXene, respectively. "Natural" in Supplementary Fig. 23b has been revised to "Neutral medium". Meanwhile, we also carefully check the correspondence between samples and data in the manuscript. We thank you again for your positive comments and valuable suggestions to improve the quality of our manuscript.

Supplementary Fig. 7. Zeta spectra of MXene, CoC_2O_4 , and $\text{CoC}_2\text{O}_4@\text{MXene}$, respectively.

Supplementary Fig. 23. b) XRD patterns of CoC₂O₄@MXene after neutral HER.

At last, we wish to thank the Reviewers and the Editors again for the constructive comments and suggestions to improve the quality of our manuscript. Thank you very much!